# Wasserstein Motifs: Non-deterministic Alignment of Ecological Networks

**Yifan Xu**
Department of Computational Applied
Mathematics and Operations Research
Rice University
Houston, TX 77005, USA
`yx76@rice.edu`

**Carlos A. Taveras**
Department of Electrical and Computer Engineering
Rice University
Houston, TX 77005, USA
`cat18@rice.edu`

**Lydia Beaudrot**
Department of Integrative Biology
Michigan State University
East Lansing, MI 48824, USA
`beaudrot@msu.edu`

**César A. Uribe**
Department of Electrical and Computer Engineering
Rice University
Houston, TX 77005, USA
`cauribe@rice.edu`

## Abstract

We study the problem of ecological network (food web) alignment, where we seek to identify structural equivalences among species and uncover backbones of interactions that represent shared functional substructures. These fundamental properties reveal the functional relationships that sustain ecosystems, enabling more accurate predictions of biodiversity responses to environmental change. Existing methods are computationally expensive, not scalable, and hard to interpret ecologically. We provide a first rigorous formalization of food web alignment based on network motifs, and show existing methods popularized in the ecological community are equivalent to minimizing a Fused Gromov-Wasserstein-like cost functional, termed *Wasserstein Motifs*. Moreover, we propose an interpretable and provably correct algorithm that efficiently computes non-deterministic alignments between food webs by leveraging their representation as feature measure networks. As a byproduct, we introduce a novel approach for identifying non-deterministic backbones of interactions in food webs. Experiments on a continental-scale dataset of 129 Sub-Saharan African mammal food webs demonstrate significant gains in accuracy, scalability, and interpretability over state-of-the-art methods. Our results establish a principled bridge between ecological network science and optimal transport, opening avenues for the analysis of complex ecological structured data.

## 1 Introduction

Ecological networks represent the web of interactions (e.g., predation) among species within an ecosystem. A species' ecological role is thus encoded in its network position, and species with similar functions tend to occupy structurally comparable niches within the network (Mora et al., 2018a). Network alignment (NA) aims to automatically identify such similarities between nodes across different ecosystems using a task-specific cost function. NA originates from the quadratic assignment problem (QAP) (Koopmans & Beckmann, 1957), which seeks a bijection between node sets that minimizes a non-convex, structure-dependent cost. Classical methods for approximating QAP solutions, such as branch-and-bound and linear relaxations, face severe scalability limitations on large graphs (Burkard, 1984). To overcome these computational barriers, modern approaches map nodes into structure-preserving embeddings, usually reducing NA to tractable linear assignment problems. The central challenge then lies in constructing effective embeddings, which are typically obtained from spectral properties of graph representations (e.g., adjacency or Laplacian matrices) (Goyal & Ferrara, 2018), or learned through models such as graph neural networks (GNNs) (Xu et al., 2019; He et al., 2024).

In Mora et al. (2018a), the authors pioneered NA for ecological networks, providing a formulation based on *network motif distributions* (small, recurring subgraphs) to quantify species similarity and alignment quality. *Network motifs* (Holland & Leinhardt, 1976; Milo et al., 2002) have been adopted widely across the machine learning and data mining communities as local building blocks for complex networks (Benson et al., 2016; Ribeiro et al., 2017; Sankar et al., 2020; Chen et al., 2021; Yu & Gao, 2022). In particular, motif counts and motif-based embeddings have been used as features for graph classification, role discovery, and representation learning, offering a compact, interpretable approach to capture higher-order connectivity beyond pairwise edges. In ecological network analysis, motifs are widely used to capture *local structural information* of food webs, mutualistic systems, and multi-trophic interaction networks (Baiser et al., 2016; Tavella et al., 2022; Cenci et al., 2018; Paulau et al., 2015). Motif structure has been linked to interpretable functional roles of species (e.g., basal producers, intermediate consumers, top predators) and to key ecosystem-level properties such as stability, robustness, and energy flow. Aggregating motif role occurrences for species yields their motif role profiles (Stouffer et al., 2012), which quantify the functional similarity of species across networks.

Mora et al. (2018a) looks for *deterministic alignments, i.e., one-to-one correspondences*. Moreover, simulated annealing is used for computations, which is slow for large networks and can lead to inconsistent alignments, especially with functionally equivalent species. An alternative motif-based approach is proposed in Almulhim et al. (2019), but it also suffers from the rigidity of deterministic alignments. While such mappings are intuitive and interpretable, they impose rigid constraints that limit ecological applicability. *In practice, species frequently exhibit overlapping or redundant functional roles (Stouffer et al., 2012).* Enforcing a strict one-to-one correspondence discards these valid alternatives and obscures functional redundancy. These drawbacks restrict the feasibility of deterministic approaches for large or heterogeneous ecological datasets.

Optimal transport (OT)- based methods (Cuturi, 2013; Sturm, 2006; Mémoli, 2011) offer superior speed compared to classical methods and enable additional modeling capabilities for NA, as alignments can be non-deterministic (many-to-many). Moreover, the recently proposed Partial Gromov-Wasserstein (PGW) distance (Bai et al., 2024) enables partial non-deterministic alignments, in which only a subset of nodes needs to be aligned. In ecological NA domains, partial non-deterministic alignments are crucial for identifying functionally equivalent species, i.e., species that play similar roles within a network. In Hung et al. (2025), NA is formulated using the Gromov-Wasserstein (GW) distance, allowing for non-deterministic alignments, but the mass conservation constraint imposed by GW lacks the flexibility to allow species playing unique functional roles to self-align, potentially forcing undesired alignments. This mass conservation constraint is relaxed in the Partial Gromov-Wasserstein distance (Bai et al., 2024) and has been applied for NA (Liu et al., 2020), but the Gromov-Wasserstein cost depends only on nodes' local neighborhood, which is insufficient for characterizing functional species equivalence. This can be mitigated by endowing nodes with a motif-based node embedding and then using a partial version of the Fused Gromov-Wasserstein distance (FGW) (Vayer et al., 2020; Chapel et al., 2020), but the quadratic cost in FGW still depends only on local connectivity.

In addition to computational modeling limitations, ecological network data are scarce because collection is time- and resource-intensive, and inconsistencies across research groups further limit the usable datasets (de Aguiar et al., 2019). Therefore, recent deep learning approaches to network alignment (Xu et al., 2019; Ratnayaka et al., 2024) have limited applicability, as the lack of large-scale, homogeneous data hinders their effectiveness for the ecological network alignment task. See Appendix A.1 for a table that summarizes recent network alignment methods.

In this paper, we leverage a motif-based ecological network representation and propose a provably correct algorithm for computing *non-deterministic motif-based* alignments between networks (see Fig. 1). A schematic overview of the complete pipeline is provided in Appendix A.3.

*Our main* **contributions** *are as follows:*
1. *Mathematical formalization of ecological network alignment:* We provide a first mathematically rigorous formulation for ecological network alignment based on network motifs.
2. *Non-deterministic alignment:* We propose a provably correct algorithm for the computation of non-deterministic food web alignments and the identification of structurally equivalent species.
3. *Backbone identification:* We introduce the notion of the backbone of interactions for non-deterministic alignments.

4. *Numerical Analysis:* We verify that our proposed formulation outperforms baselines in the ecological network alignment task and produces valid backbones on a continental-scale dataset.

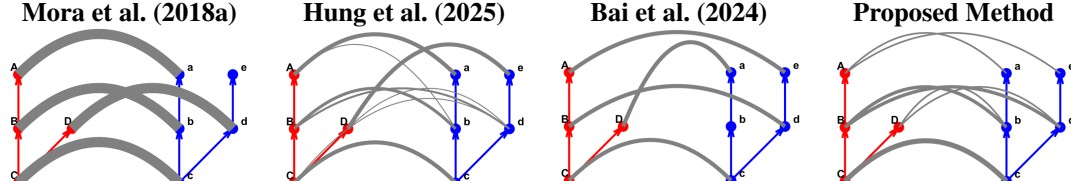

Figure 1: Alignment between two toy food webs (red: species A–D, blue: species a–e), with arrows indicating predator→prey interactions. Thicker lines indicate stronger alignment. Prior approaches (Mora et al., 2018a; Hung et al., 2025; Bai et al., 2024) produce several inconsistencies: for example, species D is often misaligned with nodes of different trophic roles, and species e is either weakly or spuriously aligned. In contrast, our method consistently aligns A–C with a–c, preserving the top-to-bottom trophic ordering, while correctly distinguishing between the structurally distinct species D and e. This results in stronger, ecologically coherent correspondences that better capture the functional roles of species across the two networks. See Appendix A.2 for more examples.

**Notations:** Denote $\mathbb{R}$ and $\mathbb{R}_+$ as the set of real and non-negative real numbers, respectively, and $[n] := \{1, 2, \cdots, n\}$ be the set of positive integers up to and including $n$. The cardinality of a set $S$ is denoted as $|S|$, and the vectors of ones in $\mathbb{R}^d$ by $1_d$. Denote the $d$-simplex by $\Delta^{d-1} = \{x \in \mathbb{R}^d_+ \mid x^\top 1_d = 1\}$, and the uniform discrete measure over the discrete set $S$ as $\nu_S$, i.e., $\nu_S := \sum_{s \in S} \delta_s$, where $\delta_s(x) = 1$ if $x = s$ and 0 otherwise. Let $\hat{\mu}$ denote a finite measure supported by a finite and discrete set $S$. We denote the probability measure induced by $\hat{\mu}$ to be $\mu$, i.e., $\mu : S \to \Delta^{|S|-1}$ is defined as $\mu := \frac{1}{\hat{\mu}(S)} \sum_{s \in S} \hat{\mu}(\{s\}) \delta_x$. The element at the $i$-th column and $j$-th row of a matrix $A \in \mathbb{R}^{m \times n}$ is denoted as $A_{ij}$ for all $i \in [m]$ and $j \in [n]$. Assuming an order for the elements in a set $S = \{s_1, \cdots, s_{|S|}\}$, we can identify the measure $\mu_S$ on $S$ with a vector in $\mu \in \mathbb{R}^{|S|}$, where $\mu_i = \mu(\{s_i\})$. The Hadamard product of two matrices $A, B \in \mathbb{R}^{m \times n}$ is denoted as $A \odot B$. The indicator function of a set is defined as $\mathbb{I}_{\mathcal{C}}(x) = 0$ if $x \in \mathcal{C}$, and $\mathbb{I}_{\mathcal{C}}(x) = +\infty$, otherwise. The diagonal matrix formed by a vector $x \in \mathbb{R}^n$ is denoted as $\text{diag}(x) \in \mathbb{R}^{n \times n}$. The number of vertices of a (measure or feature measure) network $\mathcal{X}$ is denoted as $m := |X|$. A finite, unweighted, undirected, (feature) measure network is defined as a tuple $\mathcal{X} = (X, A, \hat{\mu}, \phi)$ consisting of a set of nodes $X = \{x^1, \cdots, x^m\}$, an adjacency matrix $A \in \mathbb{R}^{m \times m}$, a fully-supported measure $\hat{\mu} : X \to \mathbb{R}^m_+$, and a feature vector $\phi : X \to \mathbb{R}^p$.

## 2 NON-DETERMINISTIC FEATURED MEASURE NETWORK ALIGNMENT

Given two featured undirected measure networks $\mathcal{X}_1 = (X_1, A_1, \hat{\mu}_1, \phi_1)$ and $\mathcal{X}_2 = (X_2, A_2, \hat{\mu}_2, \phi_2)$, we seek an alignment matrix $T \in \mathbb{R}^{m \times n}_+$ that balances structural consistency with feature similarity. Inspired by the Fused Gromov–Wasserstein (FGW) loss function (Vayer et al., 2020), we consider the following optimization problem.

$$\min_{T \in \mathcal{T}(\mathcal{X}_1, \mathcal{X}_2)} g(T; C, \alpha, \epsilon) \triangleq \underbrace{(1-\alpha)\langle T, C \rangle}_{\text{zeroth-order similarity}} + \underbrace{\alpha \langle T, A_1(T \odot C)A_2 \rangle}_{\text{first-order similarity}} + \underbrace{\Xi_{\alpha, \epsilon}(T)}_{\text{self-alignment penalty}} \quad (1)$$

$$\Xi_{\alpha, \epsilon}(T) \triangleq -\epsilon \big( \alpha \langle T, A_1 T A_2 \rangle + (1-\alpha) \|T\|_1 \big),$$

$$\mathcal{T}(\mathcal{X}_1, \mathcal{X}_2) \triangleq \{T \in [0,1]^{m \times n} \mid T1_n \preceq \hat{\mu}_1, \ T^\top 1_m \preceq \hat{\mu}_2\}.$$

where $\alpha \in [0, 1]$ is the tradeoff parameter and $\epsilon > 0$ is the self-penalty parameter. Moreover, for $x^i \in \mathcal{X}_1$ and $x^j \in \mathcal{X}_2$, we define their feature dissimilarity by $d(\phi_1(x^i), \phi_2(x^j))$, where $d : \mathbb{R}^p \times \mathbb{R}^p \to \mathbb{R}_+$, where $d$ is any nonnegative discrepancy. Collecting all pairwise dissimilarities yields the cost matrix $C \in \mathbb{R}^{m \times n}_+$ with entries $C_{ij} := d(\phi_1(x^i), \phi_2(x^j))$, where $i \in [m], j \in [n]$.

The zeroth-order similarity term promotes direct feature matching; it is minimized when the alignment assigns values proportional to the cost matrix $C$. The first-order similarity term encourages the preservation of adjacency relations under the alignment; it is minimized when the alignment matrix assigns values to entries whose neighbors are well aligned. They form a "fused" objective that

considers both vertex feature representations and network topology between vertices. The self-alignment term penalizes trivial solutions where a node is not aligned with any other node. Specifically, for an $\epsilon > 0$, we can define $C_\epsilon := C - \epsilon 1_m 1_n^\top$, and the objective in equation 1 becomes:

$$g(T; C, \alpha, \epsilon) = \alpha \langle T, A_1(T \odot C) A_2 \rangle + (1 - \alpha)\langle T, C \rangle - \epsilon(\alpha \langle T, A_1 T A_2 \rangle + (1 - \alpha)\|T\|_1).$$

Additionally, $\Xi_{\alpha,\epsilon}(T) \propto -\epsilon(\alpha \langle T, A_1 T A_2 \rangle + (1 - \alpha)(\|t_1\|_1 + \|t_2\|_1))$, where the vectors $t_1 \triangleq \hat{\mu}_1 - T1_n$ and $t_2 \triangleq \hat{\mu}_2 - T^\top 1_m$, are called the self-alignment vectors.

*Remark* 2.1. The cost matrix $C$ denotes the base feature dissimilarity matrix (non-negative by construction), while $C_\epsilon$ is a shifted version introduced solely within the optimization problem to enforce the self-alignment regularization term. Hence, although $C$ may contain only non-negative entries, $C_\epsilon$ can include negative values without affecting the interpretation of $C$ itself as a cost matrix, i.e., quantify the mass residues.

If we let $\hat{\mu}_1$ and $\hat{\mu}_2$ be probability measures, we call a solution of equation 1 an optimal *non-deterministic* alignment, or an optimal *deterministic* alignment if $T \in \{0,1\}^{m \times n}$. Informally, we distinguish whether each unit of mass (or measure) in one network is mapped exclusively to a single counterpart or may be split across multiple counterparts. Deterministic alignments (with probability measures) are thus a special case of non-deterministic alignments where $T$ is restricted to be binary. This distinction mirrors the classical relationship between the Monge and Kantorovich formulations of optimal transport (Villani et al., 2008; Kantorovich, 1942); we relax deterministic pairings to many-to-many alignments (Peyré et al., 2019). Feasible deterministic alignments correspond to a matching in a complete bipartite graph when $\alpha = 0$, which can be solved in polynomial time (Kuhn, 1955) (See Proposition 2.2). However, once structural terms are incorporated ($\alpha > 0$), the objective no longer admits such a reduction, and the problem becomes a general quadratic assignment problem.

**Proposition 2.2.** *Let $\mathcal{X}_1 = (X_1, A_1, \nu_{X_1})$, $\mathcal{X}_2 = (X_2, A_2, \nu_{X_2})$ be measure networks with uniform discrete measures over their respective vertex sets. Let $K_{m,n} = (X_1 \cup X_2, E)$ be the complete bipartite graph over their vertex sets. Then, the set of matchings in $K_{m,n}$ is in bijection with the set of deterministic alignments between $\mathcal{X}_1$ and $\mathcal{X}_2$.*

## 2.1 Computation of Non-deterministic Alignments

From an optimization perspective, the deterministic counterpart of Problem equation 1 is a nonconvex constrained optimization over transport matrices with inequality marginals. It can be classified as a combinatorial quadratic assignment problem (Koopmans & Beckmann, 1957), where approximate solutions can be efficiently computed via Gurobi Optimization, LLC (2024) and Shi et al. (2025).

In this paper, we focus on the *non-deterministic* alignment problem and propose a KL Bregman Alternating Projected Gradient (KL-BAPG) scheme with Dykstra projections. This approach preserves non-negativity by design, provides an effective trade-off between accuracy and efficiency in practice, and admits convergence guarantees under standard assumptions for Bregman alternating methods (Benamou et al., 2015; Li et al., 2023).

Given featured measure networks $\mathcal{X}_1$ and $\mathcal{X}_2$, we compute an optimal non-deterministic alignment by the following alternating KL-Bregman projected-gradient iterations (derivation in Appendix A.5):

$$\hat{T}^{(k)} = P_{\mathcal{C}_1}\big(T^{(k)} \odot \exp\big(-\gamma_k Q^{(k)}\big)\big), \quad T^{(k+1)} = P_{\mathcal{C}_2}\big(\hat{T}^{(k)} \odot \exp\big(-\gamma_k Q^{'(k)}\big)\big), \tag{2a}$$

$$Q^{(k)} \triangleq \alpha A_1(C_\epsilon \odot T^{(k)}) A_2 + \tfrac{1}{2}(1-\alpha)C_\epsilon, \quad Q^{'(k)} \triangleq \alpha C_\epsilon \odot (A_1 \hat{T}^{(k)} A_2) + \tfrac{1}{2}(1-\alpha)C_\epsilon, \tag{2b}$$

and $\mathcal{C}_1(\mu) := \{T \in [0,1]^{m \times n} \mid T1_n \preceq \mu\}$, $\mathcal{C}_2(\nu) := \{T \in [0,1]^{m \times n} \mid T^\top 1_m \preceq \nu\}$, and $\{\gamma_k\}$ is a step size sequence.

Our convergence analysis follows the KL-Bregman alternating framework (Benamou et al., 2015; Li et al., 2023), which constructs a global Lyapunov function for the two-block scheme. For the entropic generator $h(X) = \sum_{ij} x_{ij} \log x_{ij}$, the Bregman divergence $D_h$ is not symmetric, and the Lyapunov decrease holds up to an *asymmetry defect* that must be summable.

**Assumption 2.3** (Bounded accumulative asymmetrical error (AAE)). Let $h(X) = \sum_{ij} x_{ij} \log x_{ij}$ and $D_h$ be its Bregman divergence.

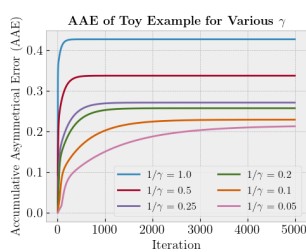

Figure 2: Cumulative asymmetry defect $\sum_{t=0}^{k} |\Delta_t|$ for constant step sizes $\gamma$.

Define $\Delta_k := D_h(\hat{T}^{(k)}, T^{(k+1)}) - D_h(T^{(k+1)}, \hat{T}^{(k)})$. Assume $\sum_{k=0}^{\infty} |\Delta_k| < \infty$.

Importantly, Assumption 2.3 is *automatically* satisfied for the entropic generator under a standard diminishing step-size regime. In Theorem C.5, we show that if $0 < \gamma_k \leq \bar{\gamma}$ and $\sum_{k=0}^{\infty} \gamma_k^3 < \infty$, then $\sum_{k=0}^{\infty} |\Delta_k| < \infty$. For computational efficiency, we also use a constant $\gamma$ in experiments and empirically monitor the cumulative asymmetry, which stabilizes across the tested step sizes (Fig. 2). Next, we state our main convergence results.

**Theorem 2.4** (Convergence to stationary points). *Let $\mathcal{X}_1 = (X_1, A_1, \mu)$ and $\mathcal{X}_2 = (X_2, A_2, \nu)$ be two featured measure networks, $C \in \mathbb{R}_+^{m \times n}$, $\alpha \in [0, 1]$, and $\epsilon > 0$. Initialize $T^{(0)} = \frac{1}{mn} 1_m 1_n^\top$ and generate $\{T^{(k)}\}_{k \geq 0}$ by equation 2. If Assumption 2.3 holds, then every accumulation point of $\{T^{(k)}\}$ is a first-order stationary point of equation 1.*

## 2.2 NETWORK MOTIFS AND ECOLOGICAL NETWORKS

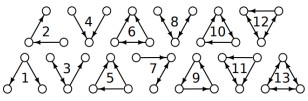

Figure 3: The 13 three-node directed motifs (adapted from (Mora et al., 2018b)).

Ecological networks are represented by directed graphs $\mathcal{G} = (X, \tilde{A})$, where $X$ is the set of species and $\tilde{A}$ is the adjacency matrix of the directed graph (not necessarily symmetric). We model them as featured measure networks $\mathcal{X} = (X, A, \mu, \mathfrak{m})$, where $A := \tilde{A} + \tilde{A}^\top$ is the adjacency matrix of the underlying undirected graph of $\mathcal{G}$, $\mu$ is a measure reflecting the relative importance of species, and $\mathfrak{m}$ is the corresponding motif role profile for this network. Specifically, given $\mathcal{G}$, its *motif role profile* is a map $\mathfrak{m}_\mathcal{G} : X \to \mathbb{R}^p$, where $(\mathfrak{m}_\mathcal{G}(x))_i$ is the number of occurrences of motif role $i$ that node $x$ participates in $\mathcal{G}$ for $i = 1, \ldots, p$.

Directedness of the interactions is captured in the motif profiles, rather than in the adjacency matrix. Although the framework supports directed adjacencies, empirical results indicate worse alignment coherence and quality (see Appendix A.6). The feature vector $\mathfrak{m}$ is defined for all directed unweighted networks; hence, its definition can carry over to arbitrary directed unweighted measure networks. Moreover, note that we decouple the directed graph into its (undirected) topological structure, captured by both $A$ and $\mathfrak{m}$, and the direction information, captured by $\mathfrak{m}$. In the rest of this paper, we will refer to this motif role profile map as $\mathfrak{m}$ when the domain of the map can be inferred. Figure 3 depicts the thirteen non-isomorphic three-node motifs for directed graphs, for a total of 30 unique role positions.

## 3 EXPERIMENTS ON LARGE-SCALE SUB-SAHARAN MAMMAL FOOD WEBS

In this section, we demonstrate the interpretability, scalability, and versatility of the proposed framework. We use Python for the following experiments, using the `NetworkX` library by Hagberg et al. (2008) for network data handling and the `pymfinder` package by Mora et al. (2018b) for network motif computations. Codes for entire pipeline on synthetic ecological networks are available in Code (2026)

**Real-World Data:** We evaluate on a *unique, real-world* dataset of 129 Sub-Saharan African mammal food webs (Hung et al., 2025). Our dataset is one of the *largest datasets of its kind* currently available, covering prey–predator interactions among 216 large-bodied terrestrial mammals ($\geq$500g), and was compiled using a metaweb approach (Dunne, 2006). The breadth of biomes and geographic coverage (Fig. 4) makes this dataset a rare opportunity to test alignment and backbone discovery at a continental scale. See Appendix A.7 for more details.

At continental scales and across many-site, multi-trophic food webs, standardized ground-truth annotations for species' functional roles are not available. This is a structural limitation of current ecological data rather than a property of our method. As a result, large-scale ecological network studies routinely rely on indirect validation (Pellissier et al., 2018; Brose et al., 2019). We follow this established practice and therefore evaluate alignments and inferred backbones using multiple complementary, dataset-level criteria rather than against unavailable ground truth.

We model all food webs as $\mathcal{G} = (X, \tilde{A})$ as featured measure networks $\mathcal{X} = (X, A, \nu_X, \mathfrak{m})$, where $A = \tilde{A} + \tilde{A}^\top$, $\nu_X$ is the uniform discrete measure over the vertex set $X$, and $\mathfrak{m}$ denotes the motif profile map. We first examine the food web alignments generated by various methods

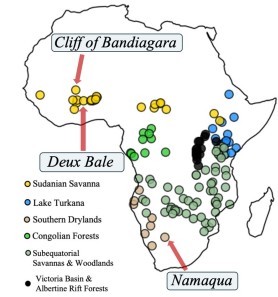

Figure 4: Locations of the 129 sites across Africa. The locations used in Fig. 5 are highlighted.

between the sites *Cliff of Bandiagara* and *Namaqua*. See Appendix A.8 for a visualization of these food webs. However, note that we compare against the FGW method by Hung et al. (2025) and the Partial-FGW method by Chapel et al. (2020) that require additional feature information about the species. The feature dissimilarity required by those methods is computed as the Gower distance between the functional trait data for each pair of species.

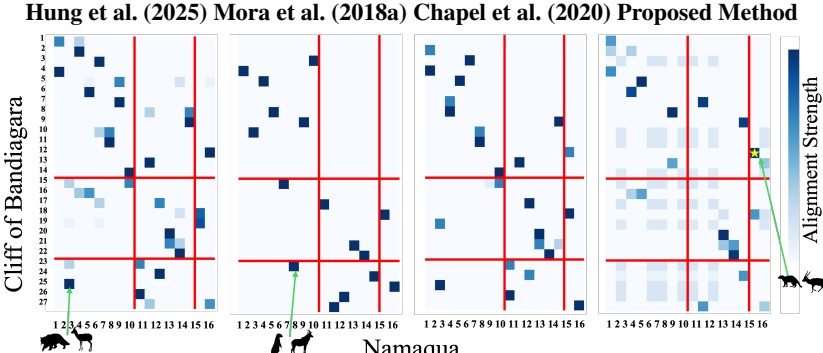

Figure 5: Alignments between Cliff of Bandiagara and Namaqua for various methods. Red lines divide species by dietary category (carnivore, omnivore, herbivore), and species are sorted by decreasing biomass within each category. See Appendix A.9 for number-species correspondences.

Figure 5 shows that the species within each dietary group are functionally similar to each other, with the exception of the alignment between *Mungos mungo* (Banded mongoose, an insectivore) with *Raphicerus campestris* (Steenbok, a herbivore), marked with a star. Upon further inspection, we find that the prey of the banded mongoose have body masses $\leq 500g$ and are therefore excluded from the network. As a result, the banded mongoose does indeed play a functional role in *Cliff of Bandiagara* that is similar to that of the steenbok, testifying to the reliability of the proposed method. However, the exceptions identified in the FGW and Partial-FGW alignments are less ecologically consistent. For example, both of these methods aligned *Felis silvestris* (Wildcat, a carnivore) to *Eudorcas rufifrons* (Red fronted gazelle, a herbivore). Unlike the prior case, the wildcat feeds on small mammals, four of which are also present in the *Namaqua* food web. On the contrary, the red fronted gazelle is near the bottom of the food chains in *Cliff of Bandiagara*, only very rarely feeding on *Ichneumia albicauda*, the white tailed mongoose. The wildcat and the gazelle play different functional roles in their ecological communities, yet are strongly aligned by the FGW and partial-FGW methods. The method in Mora et al. (2018a) produces a similar exception where the *Hippotragus equinus* (Roan antelope, a herbivore) is aligned to the *Suricata suricatta* (Meerkat, a carnivore).

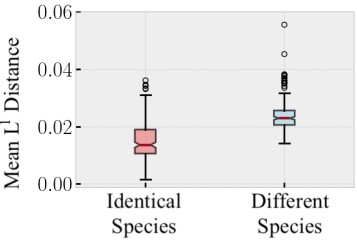

Figure 6: Alignment discrepancy.

Next, we describe the performance of the proposed method pairwise across all locations in the dataset, for a total of 8256 alignments. In our pipeline, the alignment solver itself is sufficiently fast that feature and cost construction become the dominant computational bottleneck, whereas in Mora et al. (2018a), the alignment procedure dominated runtime. At the dataset level, motif computation can be amortized across alignments, whereas prior methods derive limited benefit from amortization because alignment optimization itself dominates runtime. This shift in the computational bottleneck enables efficient dataset-level analyses involving thousands of pairwise alignments rather than isolated comparisons.

We fix $\alpha = 0.5$ as a default balance between motif-based and structural similarity, and choose $\epsilon$ from an approximation function (Eq. equation 6) to control the desired proportion of self-aligned mass. Empirically, performance varies smoothly with $\alpha$, and results are robust across a wide range of values; full sensitivity analyses are provided in Appendix A.10. In a cold-start setting, where no preprocessing is reused, total end-to-end runtime includes motif enumeration, cost-matrix construction, and alignment. In this regime, aligning two food webs with up to 60 species takes $0.20 \pm 0.08$ seconds on an AMD Ryzen 7 9700X CPU, with most of the time spent on motif and cost-matrix construction. This represents a 40x speedup over the 7.92 seconds cold-start average

of the method in Mora et al. (2018a), with the same motif enumeration routines and in the same computing environment.

Figure 6 shows that our alignment method captures species' functional roles. We adopt the common assumption that identical species across different communities typically play more similar roles than arbitrary non-identical species. As a validity check to test whether our alignments are consistent with this assumption, we defined the alignment discrepancy between two species as the average $L_1$ difference of their alignment distributions across networks. We then compared discrepancies between identical and non-identical species. *Results show that identical species exhibited a mean discrepancy of* $0.015 \pm 0.008$*, significantly lower than the* $0.024 \pm 0.007$ *observed for non-identical species.* These results demonstrate that the proposed method yields significantly more consistent alignments for identical species, verifying its ability to recover functional role similarities. Other established OT-based baselines exhibit this behavioral trend as well (Appendix A.4).

Figure 7 shows the meta alignment obtained by partitioning the 216 species into six groups based on dietary class and median biomass. Each group is treated as a supernode, with alignment strengths computed as weighted averages of species-level alignments. For each pair of species and the alignment between them, the weight is defined as the product of the sizes of the two networks from which the two species are derived, so that the alignment strengths are comparable across networks. Figure 7 also shows strong alignment within large and small carnivores, but a weak alignment between the two, consistent with the exclusion of small-prey species ($\leq 500g$) that lowers the trophic placement of small carnivores relative to larger ones. The baseline methods that do not take species dietary categories as input consistently produce weaker group-level coherence and less consistent structure than our method (Appendix A.4). To further assess the robustness of our approach, we present results on alignments on additional real and synthetic data in Appendix A.11.

## 4 IDENTIFICATION OF NON-DETERMINISTIC BACKBONES OF INTERACTIONS

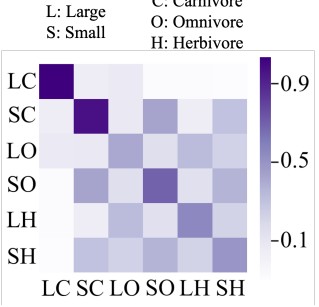

Figure 7: Weighted average of alignment strengths across and within groups of species.

*Backbones of interactions* identify recurrent structural patterns in food webs that persist despite ecological or environmental variability. These backbones highlight trophic links that are consistently conserved across different ecosystems, thereby revealing ecological regularities that shape community assembly and constrain possible reorganization under disturbance. The existing approach to backbone identification relies on deterministic alignments, in which species are mapped one-to-one across networks (Mora et al., 2018a). However, the rigidity of deterministic methods is also evident in backbone identification. By recovering the full set of valid role correspondences within each pairwise alignment of ecological networks, non-deterministic alignments produce backbones that better reflect structural regularities across ecosystems. Formally, backbones are defined by role-similarity scores that summarize how strongly a species aligns with its counterparts across a dataset of ecological networks. Species with consistently high role similarity scores are considered central to the structural backbone, and the induced subgraph formed by these species provides a rigorous representation of the backbones. We now introduce the relevant definitions.

**Definition 4.1.** Let $\mathcal{D} = \{\mathcal{X}_1, \ldots, \mathcal{X}_N\}$ denote a collection of featured measure networks, and let $\{T^{ij}\}_{i,j=1}^N$ be the pairwise non-deterministic alignments between them with corresponding dissimilarity matrices $\{C^{ij}\}_{i,j=1}^N$. For a species $u \in X_i$, its role similarity score $\mathcal{S}$ is defined as $\mathcal{S}(u) = \sum_{p=1}^N \sum_{v \in X_p} (1 - C_{uv}^{ip}) T_{uv}^{ip}$. A species achieves a high role similarity score when little of its mass is self-aligned, and it aligns with species that have a low dissimilarity cost across the dataset.

We assume that the role dissimilarity $C$ is obtained from a similarity score $\rho \in [0, 1]$ as $C = 1 - \rho$; this holds for Wasserstein Motifs. More generally, any bounded role cost can be normalized to this form without affecting the subsequent construction.

**Definition 4.2** (Top-$k$ backbone). Given a network $\mathcal{X}_i$ and the role similarity scores of its species, the *top-k backbone* of $\mathcal{X}_i$, denoted $B_i = (V_{B_i}, E_{B_i})$, is the subgraph induced by the $k$ species in $X_i$ with the highest role similarity scores.

We adopt the criteria of Mora et al. (2018a) to assess the internal consistency and ecological coherence of $B_i$: (i) *relative connectance*, the ratio of connectance within the backbone compared to the full

network; (ii) *connectivity* ("path likelihood" in Mora et al. (2018a)), the fraction of backbones that induce a connected subnetwork; and (iii) *transitivity*. While (i) and (ii) lift directly to our setting, (iii) requires a new definition because our proposed correspondences are fractional. Therefore, we introduce the following notion of non-deterministic transitivity:

**Definition 4.3.** For each species $u \in V_{B_i}$, its *non-deterministic transitivity score* is

$$\mathrm{Tr}(u) \triangleq \sum_{\substack{p,q\in[N] \\ p<q, p,q\neq i \\ v\in X_p, w\in X_q}} T_{uv}^{ip} \cdot T_{uw}^{iq} \cdot T_{vw}^{pq} \bigg/ \sum_{\substack{p,q\in[N] \\ p<q, p,q\neq i \\ v\in X_p, w\in X_q}} T_{uv}^{ip} \cdot T_{uw}^{iq}.$$

If the denominator in the transitivity score is zero, we define the score to be zero. This corresponds to the case where species $u$ does not participate in any nonzero alignment triangles with other species in the dataset, and therefore exhibits no transitive alignment structure. The transitivity score of a backbone is obtained by averaging $\mathrm{Tr}(u)$ across all $u \in V_{B_i}$. It measures the consistency with which species alignments form transitive triangles across networks.

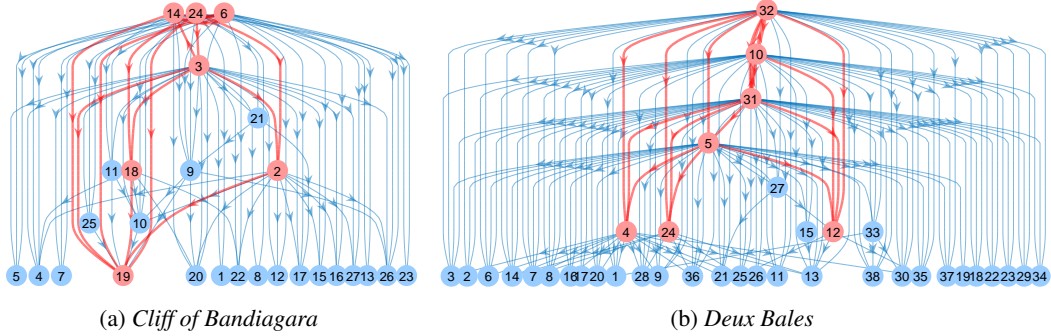

(a) *Cliff of Bandiagara*                    (b) *Deux Bales*

Figure 8: Examples of the top-7 backbones of food webs in our dataset. Species in the backbone are marked red, while the remaining species are blue. The full correspondence between number labels and species names can be found in Appendix A.9.

We empirically analyze the backbones of interactions underlying the food webs in our dataset. Figure 8 shows the top-7 backbones of two of our food webs. We first computed role similarity scores (Def. 4.1) for all species for each network, and sorted them in decreasing order. Then, for a given $k$, we compute the subnetwork induced by the top-$k$ species to obtain the top-$k$ backbone (Def. 4.2). We measure the connectivity and transitivity of the backbones we generate to assess their quality.

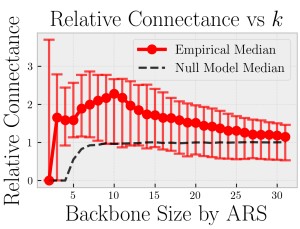 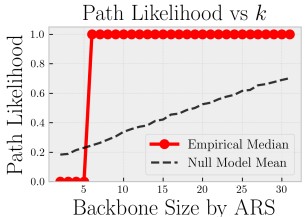 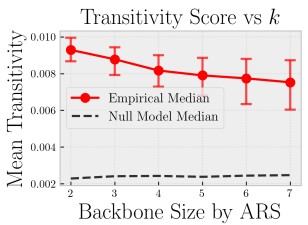

(a) Relative connectance, the ratio of backbone connectance to that of the full food web.

(b) Path likelihood, the fraction of backbones that induce a connected subnetwork.

(c) Transitivity, non-deterministic alignment transitivity score (Def. 4.3).

Figure 9: Structural properties of our top-$k$ backbones as a function of $k$.

Figure 9 presents the relative connectance, path likelihood, and transitivity score (defined in Def. 4.3) for the corresponding backbones produced by our method for each backbone size $k$. These scores are compared to a null model, in which the same computations are performed on the subnetwork induced by $k$ randomly selected species, averaged over 50 trials. Our backbones demonstrated a significantly higher relative connectance than the null model for smaller values of $k$. For large values of $k$, it is expected that the relative connectance of both our backbones and the null backbones converges to 1, because both backbones cover the vast majority of species in most food webs, making them have similar connectance to the full web. Moreover, our backbones are much more likely to form a single connected component than the null model, as demonstrated by having more than half of the backbones connected with as few as 6 species, which is significantly fewer than the 19 species required for the

null backbones. Lastly, our backbones are consistently more transitive than the null model, exhibiting high transitivity scores for all values of $k$ up to 7. Definition 4.3 involves summing over all triples of networks and all triples of backbone species, yielding a time complexity of $O(N^3 k^3)$, which is polynomial and tractable for typical ecological datasets. In practice, the transitivity evaluation is dominated by the *null-model estimation*, which requires recomputing the metric across 50 random subsets. On our desktop CPU, the entire null-model computation took approximately 20 hours, a cost incurred once solely for evaluation purposes. Overall, these findings demonstrate that our method produces backbones that consistently meet the backbone criteria proposed in Mora et al. (2018a), while also offering scalability and reproducibility. Although OT-based baselines produce reasonable backbones under the same evaluation protocol, they generally trade off connectivity and transitivity, whereas *Wasserstein Motifs* consistently yield backbones that are both connected and highly transitive (Appendix A.4).

## 5    CONCLUSION

We presented *Wasserstein Motifs*, a non-deterministic network alignment framework tailored to ecological interaction networks. Our method computes non-deterministic correspondences by matching *functional roles* rather than species identities, enabling principled comparison of food webs across locations where node labels are not directly comparable. We proposed an efficient KL-Bregman alternating scheme with convergence guarantees, and we used the resulting alignments to construct *backbones* that capture the most consistently supported interactions across networks. On a continental-scale collection of African mammal food webs, Wasserstein Motifs identifies coherent clusters of functionally similar species and downweights redundant taxa, producing backbones with substantially higher connectivity and transitivity than a matched null model. These results suggest that the method extracts stable organizational principles of ecological communities while remaining flexible to heterogeneous inputs (traits, abundances, and interaction structure).

## MEANINGFULNESS STATEMENT

This work introduces meaningful representations of life by formalizing how *functional roles*, rather than species identities, are encoded, compared, and conserved across ecosystems. Inspired by tools in optimal transport, our ecological network alignment method captures higher-order interaction patterns using network motifs and reveals trophic structure, redundancy, and functional equivalences. The resulting non-deterministic alignments uncover backbones of ecological interactions across food webs from various ecosystems, offering interpretable, efficient abstractions of ecosystem organization that align with ecological theory.

## ACKNOWLEDGEMENTS

We thank Kai M. Hung for exploratory code and insights regarding the dataset. We thank Ann E. Finneran for the early ecological conceptualization. This work is funded by the National Science Foundation under Grants #2213568 and #2443064.

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

# A APPENDIX

## A.1 TABLE OF COMPARISON

Table 1: Comparison with existing network alignment methods

| METHOD | DESIRED PROPERTY | | | |
|---|---|---|---|---|
| | NON-DET. | TOP.-AWARE | PARTIAL | MOTIF-BASED |
| Mora et al. (2018a) | ✗ | ✓ | ✓ | ✓ |
| Hung et al. (2025) | ✓ | ✓ | ✗ | ✗ |
| Bai et al. (2024) | ✓ | ✓ | ✓ | ✗ |
| Xu et al. (2019) | ✓ | ✓ | ✗ | ✗ |
| Ratnayaka et al. (2024) | ✗ | ✗ | ✓ | ✗ |
| Almulhim et al. (2019) | ✗ | ✓ | ✗ | ✓ |
| Wasserstein Motifs (Ours) | ✓ | ✓ | ✓ | ✓ |

## A.2 More Alignment Toy Examples

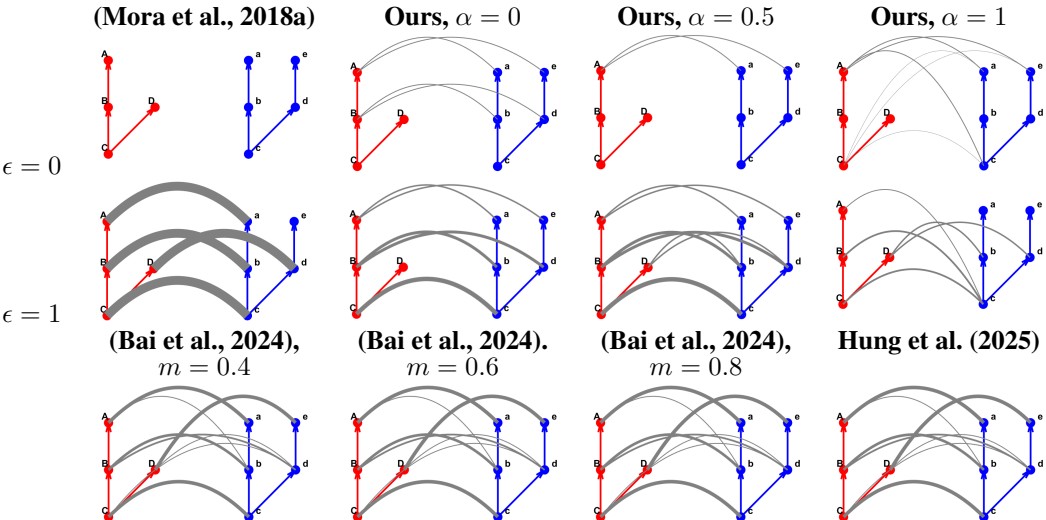

Figure 10: Optimal alignment between two toy networks with various alignment methods. Thicker lines indicate stronger alignment.

## A.3 WASSERSTEIN MOTIFS PIPELINE

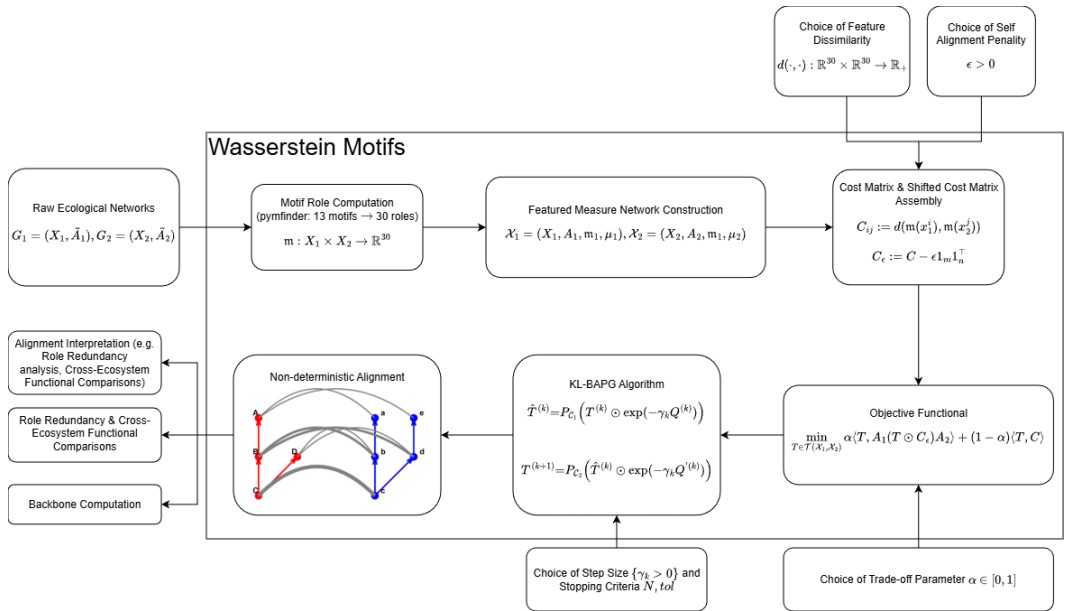

Figure 11: Pipeline Diagram for Wasserstein Motifs Framework.

### A.4 Motif-Based Fused Gromov-Wasserstein Variants

**First-order alignment discrepancy vs. Gromov-Wasserstein** We further expound on the difference between our proposed formulation 1 and that of Fused Gromov-Wasserstein (FGW). We emphasize the difference between our first-order term and its analogue in FGW (i.e., the Gromov-Wasserstein term).

Given a pair of networks $G_1$ and $G_2$, our first-order term is

$$\langle T, A_1(T \odot C)A_2 \rangle = \sum_{i,j,i',j'} (A_1)_{ii'}(A_2)_{jj'}C_{i'j'} T_{ij}T_{i'j'},$$

whereas the GW distance term is

$$\sum_{i,j,i',j'} L(d_1(x_i, x_{i'}), d_2(y_j, y_{j'})) T_{ij}T_{i'j'}.$$

The critical difference between these terms lies in our cost matrix $C$, which is comprised of distances between the features of a pair of nodes $v_i' \in G_1$ and $v_j' \in G_2$. The GW distance, on the other hand, assumes that the spaces it is computing the distance between are incomparable Mémoli (2011), implying that a cost between the vertices of $G_1$ and $G_2$ cannot appear. Therefore, there is no choice of $L$ for which our first-order term and the GW distance are equal, in general. If $L(x, y) = xy$ and $C$ is the all-ones matrix, these terms coincide, but this case is not practical, as it implies that all nodes have the same motif profile.

**Baseline Comparison.** To determine whether the improved performance of our method over Gromov-Wasserstein-based methods is due to the cost-function formulation or the feature choice, we modify the baselines by incorporating motif-role dissimilarity. In particular, we consider the following baselines, the last two of which are augmented with motif-based features/distances:

- **Vanilla FGW** Hung et al. (2025): Gower trait distance; shortest-path distance as metric (shown in Figs. 13 and 14).
- **FGW (traits as features, motifs as metric):** Gower trait distance as features; motif-role dissimilarity as metric (shown in Figs. 15 and 16).
- **FGW (motifs as features and metric):** motif-role dissimilarity used for both features and metric (shown in Figs. 17 and 18).

We compare Wasserstein Motifs to OT-based baselines as they are originally formulated, without enforcing additional normalization or post-processing. All methods operate on the same input networks and, where applicable, the same node features; however, they differ in their constraints and objectives (e.g., equality versus inequality constraints). Enforcing a common normalization would therefore alter the problems being solved and obscure the intended modeling differences. Instead, we evaluate each method under its native formulation and assess alignment quality and backbone structure through downstream ecological metrics applied consistently across methods.

All baselines were assessed against the $L_1$ alignment discrepancy (Fig. 6), meta-web alignment (Fig. 7), and backbone quality (Fig. 9) produced by our method.

All FGW variants separate identical vs. non-identical species in the discrepancy test. For meta-alignment, FGW with trait features performs well, as biomass and diet directly encode functional roles. The motif-only FGW variant performs noticeably worse for large carnivores, demonstrating that simply inserting motif features into FGW does not reproduce the expressivity of our structural term. FGW-based backbones also exhibit structural inconsistencies:

- **Vanilla FGW:** connected but **non-transitive backbones**,
- **FGW (traits as features, motifs as metric):** transitive but **largely disconnected**,
- **FGW (motifs as both features and metric): lower connectance** than Wasserstein Motifs.

Our method, in contrast, consistently produces backbones that are both connected and transitive, uniquely allowing *controllable self-alignment* via the regularizer $\epsilon$, a capability that FGW lacks. Overall, these findings demonstrate that our improvements stem from the *objective design*, not from feature choice, and that FGW/Partial-FGW baselines are neither structurally nor computationally comparable to our method.

**On Partial-FGW.** We additionally examined Partial-FGW, as it is the closest FGW-based approach conceptually aligned with our goal of non-deterministic correspondences. However, we found it to be computationally infeasible at ecological scales. In an experiment computing pairwise alignments across a representative subset of networks (spanning the full range of 20–75 species), Partial-FGW required *up to 15 seconds per alignment* for networks with $\sim 60$ nodes (as shown in Fig. 12). In contrast, FGW and our method required about $\sim 0.05$ seconds. Since our analysis requires 8256 pairwise alignments, Partial-FGW would require several days of computation and is therefore not a practical baseline for ecological network data. This is consistent with prior reports of Partial-FGW scaling poorly when structural consistency dominates the cost.

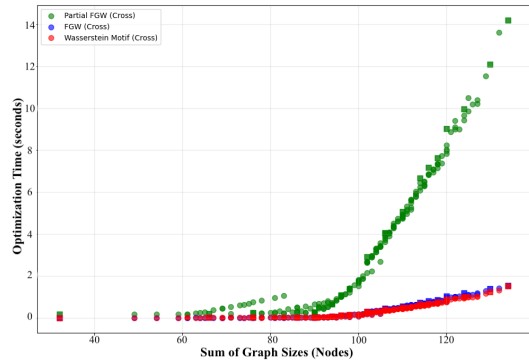

Figure 12: Optimization Time vs Graph Sizes for our method and the Baselines.

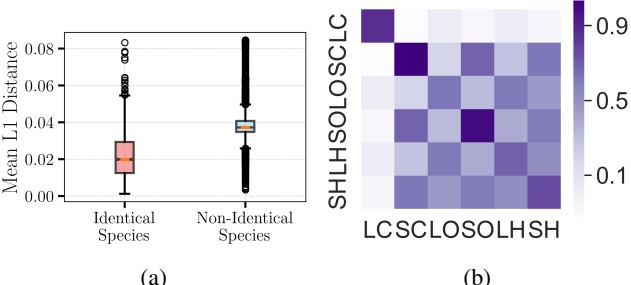

Figure 13: Alignment Discrepancy and Meta-alignment for Vanilla FGW

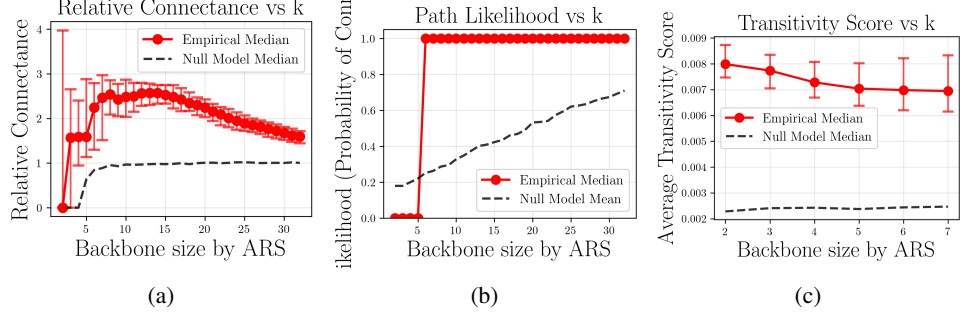

Figure 14: Statistical Analysis of Top-k backbones for Vanilla FGW

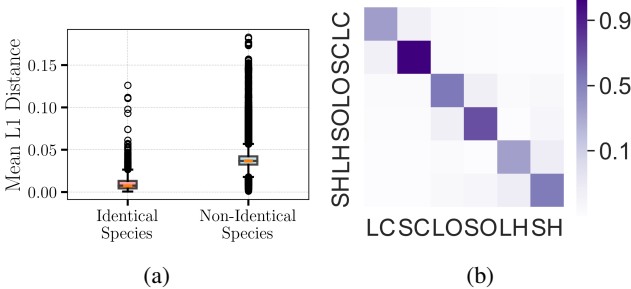

Figure 15: Alignment Discrepancy and Meta-alignment for FGW with Traits as features and motif-role dissimilarity as metric

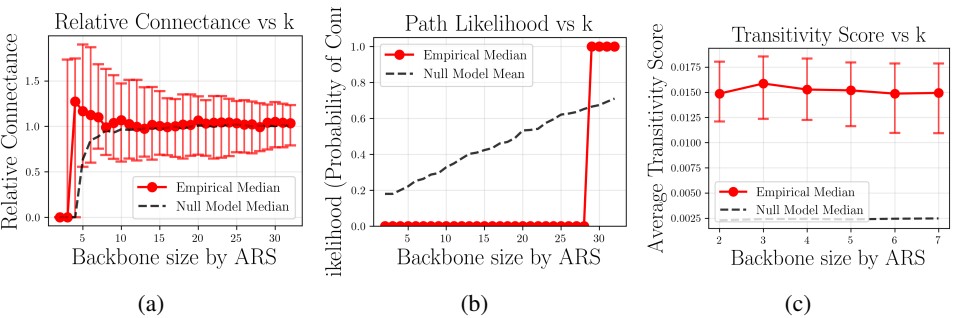

Figure 16: Statistical Analysis of Top-k backbones for FGW with Traits as features and motif-role dissimilarity as metric

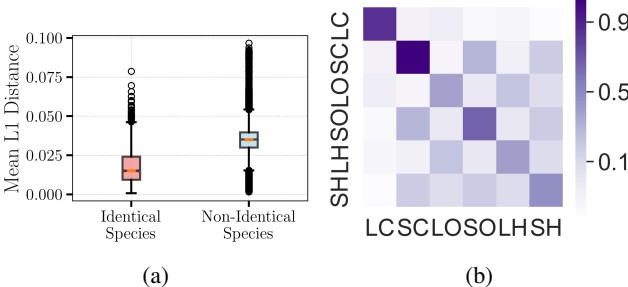

Figure 17: Alignment Discrepancy and Meta-alignment for FGW with motif-role based features and motif-role dissimilarity as metric

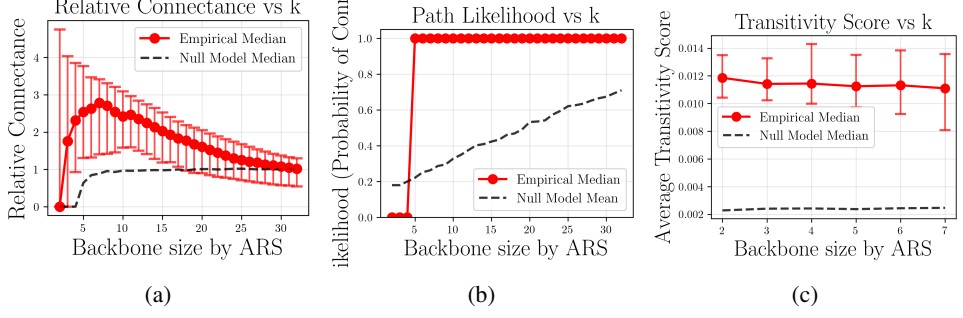

Figure 18: Statistical Analysis of Top-k backbones for FGW with motif-role based features and motif-role dissimilarity as metric

### A.5 ALGORITHM ANALYSIS AND DETAILS

We can rewrite our objective functional in Eq. equation 1 as follows:

$$g(\pi, w; C, \alpha, \epsilon) = \alpha \langle \pi, A_1(C \odot w) A_2 \rangle + (1-\alpha) \langle C, \frac{1}{2}(\pi + w) \rangle + \Xi_{\alpha,\epsilon}(\pi, w),$$

where $\Xi_{\alpha,\epsilon}(\pi, w) = -\epsilon(\alpha \langle \pi, A_1 w A_2 \rangle + \frac{1}{2}(1-\alpha)(\|\pi\|_1 + \|w\|_1))$, thus Eq. equation 1 is equivalent to

$$\min_{\pi=w} g(\pi, w; C, \alpha, \epsilon) + \mathbb{I}_{\mathcal{C}_1}(\pi) + \mathbb{I}_{\mathcal{C}_2}(w), \tag{3}$$

where $\mathbb{I}_{\mathcal{C}_i}$ denotes the indicator function for the constraint set $\mathcal{C}_i$. Then, we can penalize the equality constraint in the cost function and construct the penalized cost function:

$$\min_{\pi, w} g(\pi, w) + \mathbb{I}_{\mathcal{C}_1}(\pi) + \mathbb{I}_{\mathcal{C}_2}(w) + \frac{1}{\gamma} D_h(\pi, w), \text{ with } D_h(X, Y) \triangleq \sum_{ij} \left[ x_{ij} \log \frac{x_{ij}}{y_{ij}} \right],$$

where $D_h$ is the Bregman divergence induced by $h(X) = \sum_{ij} x_{ij} \log x_{ij}$, and $\gamma > 0$. Now, following Li et al. (2023), at iteration $k$, we alternate between linearizing $g$ around each operator ($\pi$ and $w$) and solve a KL–proximal step subproblem:

$$\pi^{k+1} = \arg \min_{\pi \in \mathcal{C}_1} \left\langle Q^{(k)}, \pi \right\rangle + \frac{1}{\gamma} D_h(\pi, w^k) \text{ and } w^{k+1} = \arg \min_{w \in \mathcal{C}_2} \left\langle Q'^{(k)}, w \right\rangle + \frac{1}{\gamma} D_h(w, \pi^{k+1}),$$

where the cost surrogates $Q^{(k)}$ and $Q'^{(k)}$ are defined as:

$$Q^{(k)} := \alpha A_1(C \odot w^k) A_2 + \frac{1}{2}(1-\alpha)C - \epsilon(A_1 w^k A_2 + \frac{1}{2} 1_m 1_n^\top)$$

$$Q'^{(k)} := \alpha C \odot (A_1 \pi^{k+1} A_2) + \frac{1}{2}(1-\alpha)C - \epsilon(A_1 \pi^{k+1} A_2 + \frac{1}{2} 1_m 1_n^\top).$$

In each of the subproblems, we minimize a linear term plus the Bregman divergence between the previous iteration of the other operator, under nonnegativity and a *single* set of marginal constraints.

We write the generic subproblem as $\min_{X \in \mathcal{C}} \langle Q, X \rangle + \frac{1}{\gamma} D_h(X, Y)$ with $\mathcal{C} \in \{\mathcal{C}_1, \mathcal{C}_2\}$. Setting $\mathcal{C} = \mathcal{C}_1$ and introducing dual variables $\lambda \in \mathbb{R}_+^m$, the Lagrangian is

$$\mathcal{L}(X, \lambda) := \langle Q, X \rangle + \frac{1}{\gamma} D_h(X, Y) + \langle \lambda, X 1_n - \mu_1 \rangle \tag{4}$$

Minimizing w.r.t. $X$ entrywise yields, for any $x_{ij} > 0$:

$$0 = \frac{\partial \mathcal{L}}{\partial X_{ij}} = Q_{ij} + \frac{1}{\gamma} \log \frac{X_{ij}}{Y_{ij}} + \lambda_i \implies X_{ij} = Y_{ij} \exp\left( -\gamma \left( Q_{ij} + \lambda_i \right) \right) \tag{5}$$

If we denote $\hat{X}_{ij} = Y_{ij} \exp(-\gamma Q_{ij})$ as the unconstrained optimizer, then the Lagrangian optimizer can be written as $X_{ij}^\star = Y_{ij} \exp(-\gamma Q_{ij}) \exp(-\gamma \lambda_i) = \hat{X}_{ij} \exp(-\gamma \lambda_i)$, with complementary slackness $\lambda_i^\star (X^\star 1_n - \mu_1)_i = 0$ for all $i \in [m]$. An analogous derivation with column constraints $\mathcal{C} = \mathcal{C}_2$ introduces $\eta \in \mathbb{R}_+^n$ and yields

$$X_{ij}'^\star = Y_{ij}' \exp\left(-\gamma Q_{ij}'\right) \exp\left(-\gamma \eta_j^\star\right), \quad \eta_j^\star \left( X^{\star\top} 1_n - \mu_2 \right)_j = 0 \quad \forall j \in [n].$$

The structure of the Lagrangian optimizer shows that enforcing $\pi 1_n \preceq \mu_1$ (or $w^\top 1_m \preceq \mu_2$) is equivalent to scaling each row $i$ (or column $j$) of the unconstrained optimizer $\hat{X}$ by a factor $\exp(-\gamma \lambda_i)$ (or $\exp(-\gamma \eta_j)$). Complementary slackness implies that:

- if the current row sum $(\hat{X} 1_n)_i \leq (\mu_1)_i$ (or column sum $(\hat{X}^\top 1_m)_j \leq (\mu_2)_j$), then $\lambda_i^* = 0$ (or $\eta_i^* = 0$) and we leave the row (or column) unchanged,
- if $(\hat{X} 1_n)_i \geq (\mu_1)_i$ (or $(\hat{X}^\top 1_m)_j \geq (\mu_2)_j$), then $\lambda_i^* > 0$ (or $\eta_j^* > 0$) and the row (or column) is uniformly scaled until it satisfies the constraint.

Therefore, the Dykstra projection steps onto the constraint sets are defined as

$$P_{\mathcal{C}_1}(\hat{X}) := \text{diag}\left( \min\left( \frac{\mu_1}{\hat{X} 1_n}, 1_m \right) \right) \hat{X}, \quad \text{and} \quad P_{\mathcal{C}_2}(\hat{X}) := \hat{X} \text{diag}\left( \min\left( \frac{\mu_2}{\hat{X}^\top 1_n}, 1_n \right) \right).$$

Algorithm 1 summarizes our algorithm.

---

**Algorithm 1:** KL-BAPG with Dykstra Projections

---

**Input** :
- featured measure networks $\mathcal{X}_1 = (X_1, A_1, \mu_1, \phi_1), \mathcal{X}_2 = (X_2, A_2, \mu_2, \phi_2)$.
- the feature discrepancy $d : \mathbb{R}^M \times \mathbb{R}^M \rightarrow \mathbb{R}_+$.
- the tradeoff parameter $\alpha \in [0, 1]$.
- the self-alignment regularization parameter $\epsilon > 0$.
- the step size $\gamma > 0$.
- stopping criteria for main loop (maximum iterations $K$, step tolerance, etc.).

**Output** : $T^\star$, an approximate solution to the non-deterministic alignment problem between $\mathcal{X}_1$ and $\mathcal{X}_2$.

1  Normalize $\mu_1$ and $\mu_2$;

2  $C_{ij} \leftarrow d(\phi_1(x^i), \phi_2(x^j)) - \epsilon$ for all $x^i \in X_1$ and $x^j \in X_2$ ;            // Uniform shift

3  $T^{(0)} \leftarrow \frac{1}{mn} 1_m 1_n^\top$ ;                        // Initialize with uniform matrix

4  **foreach** $k = 1, 2, \cdots, K$ **do**

5  $\quad$ $Q^{(k)} \leftarrow \alpha A_1 (C \odot T^{(k-1)}) A_2 + \frac{1}{2}(1-\alpha)C$;

6  $\quad$ $\hat{T}^{(k)} \leftarrow T^{(k-1)} \odot \exp\left(-\gamma Q^{(k)}\right)$ ; // Unconstrained Lagrangian optimizer

7  $\quad$ $\hat{T}^{(k)} \leftarrow \text{diag}(\min(\frac{\mu_1}{\hat{T}^{(k)} 1_n}, 1_m)) \hat{T}^{(k)}$ ;            // Proj. into row constraints

8  $\quad$ $Q^{(k)'} \leftarrow \alpha C \odot (A_1 \hat{T}^{(k)} A_2) + \frac{1}{2}(1-\alpha)C$;

9  $\quad$ $T^{(k)} \leftarrow \hat{T}^{(k)} \odot \exp(-\gamma Q^{(k)'})$ ;    // Unconstrained Lagrangian optimizer

10 $\quad$ $T^{(k)} \leftarrow T^{(k)} \text{diag}(\min(\frac{\mu_2}{T^{(k)\top} 1_m}, 1_n))$ ;   // Proj. into column constraints

11 $\quad$ Break if any stopping criterion is met;

12 **return** $T^{(k)}$;

---

A.6 DIRECTED ADJACENCY VARIANT

This section explores the variant of Wasserstein Motifs where directed adjacency is used instead of its undirected counterpart in Equation 1. Importantly, none of our theoretical results require the adjacency matrices $A_1$ and $A_2$ to be symmetric. The first-order term

$$\langle T, \; A_1(T \odot C)A_2 \rangle$$

remains well-defined for any nonnegative matrices, and directed adjacencies (e.g., separate in-/out-adjacency matrices) are therefore a drop-in replacement in our framework.

However, using directed adjacencies changes the ecological interpretation of the first-order similarity. In this variant, the structural term compares only *directed neighbors* (e.g., prey sets), whereas our main formulation averages over predator–prey interactions by symmetrizing the adjacency. This shift places greater weight on trophic directionality and less on undirected neighborhood structure.

To evaluate this directed variant, we repeated all key experiments - pairwise alignments, role-consistency analysis, meta-alignment, and backbone quality - with directed adjacency matrices. Figure 19 and 20 report the results. While the method remains capable of distinguishing identical species from non-identical ones, we observed two consistent degradations:

1. **Meta-alignment performance declined substantially.** Alignment strengths across networks became noisier, and trophic-group structure was less pronounced.
2. **Backbone quality weakened.** Although the resulting top-$k$ backbones remained connected, their non-deterministic transitivity was only slightly above the null model, which is far below the improvement observed under our undirected formulation.

Overall, while the framework technically supports directed adjacencies, our empirical analysis indicates that the directed version yields weaker alignment coherence and backbone structure. We attribute this degradation to the loss of structural information when only outgoing interactions (prey) are considered, whereas ecological role similarity is typically driven jointly by both predator and prey relationships.

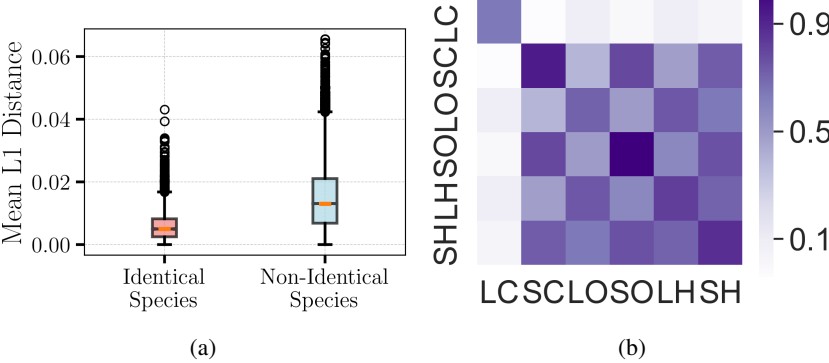

(a)             (b)

Figure 19: Alignment Discrepancy and Meta-alignment for the Directed-adjacency Variant

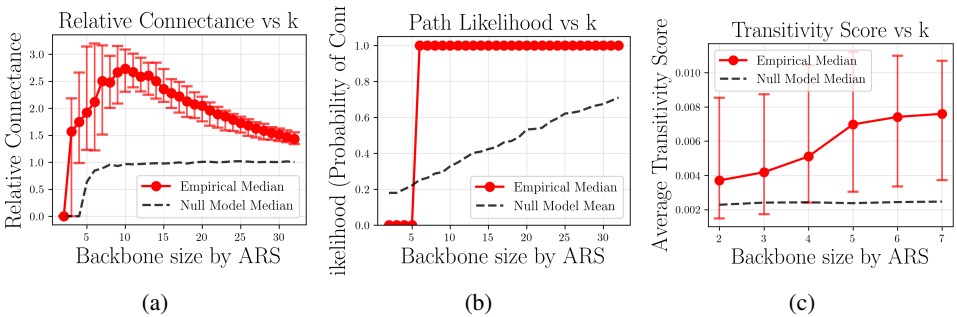

Figure 20: Statistical Analysis of Top-k backbones for the Directed-adjacency Variant

## A.7 DATASET DESCRIPTION AND STATISTICS

We use a dataset of 129 Sub-Saharan African mammal food webs (Hung et al., 2025), which contains prey–predator interactions for large-bodied terrestrial mammals ($\geq$ 500g). The dataset comprises 216 mammal species from 12 orders and 33 families. Figure 4 shows the locations of the food webs, classified by biomes. Interactions were compiled using a metaweb approach (Dunne, 2006), where a comprehensive binary interaction matrix was constructed from the union of predator–prey interactions from Mammals of Africa (Kingdon, 2013) and global mammal interaction data (Fricke et al., 2022), and additional inferred interactions based on taxonomic information and body size overlap (Brose et al., 2019; Gravel et al., 2013; Segar et al., 2020; Woodward et al., 2005). See Hung et al. (2025) for methodological details of the dataset assembly. Figure 21 provides a statistical overview of the dataset. Species richness across sites ranges from 16 to 75 and connectance from 0.0345 to 0.221 (Figs. 21a–21b). Taxonomically, the dataset spans herbivores($\approx$ 42.6%), omnivores($\approx$ 34.5%), and carnivores ($\approx$ 22.9%), with a detailed breakdown shown in Fig. 21c. While 14.8% species are restricted to a single site, a large group recurs broadly across ecosystems (Fig. 21d).

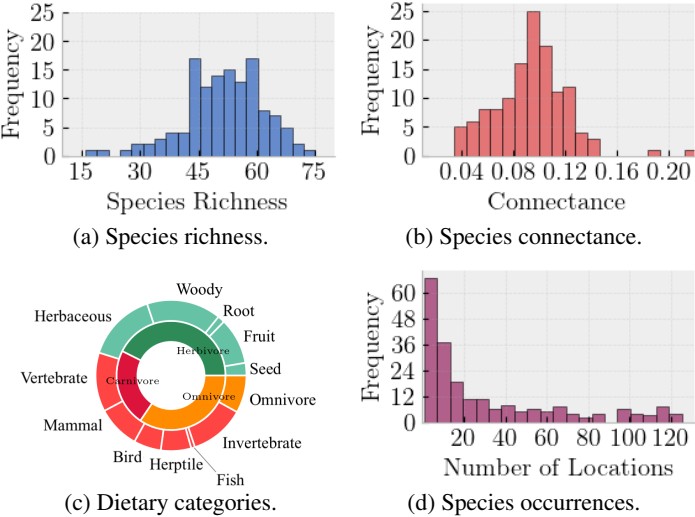

(a) Species richness.

(b) Species connectance.

(c) Dietary categories.

(d) Species occurrences.

Figure 21: Dataset Statistics

## A.8 FOOD WEB VISUALIZATION

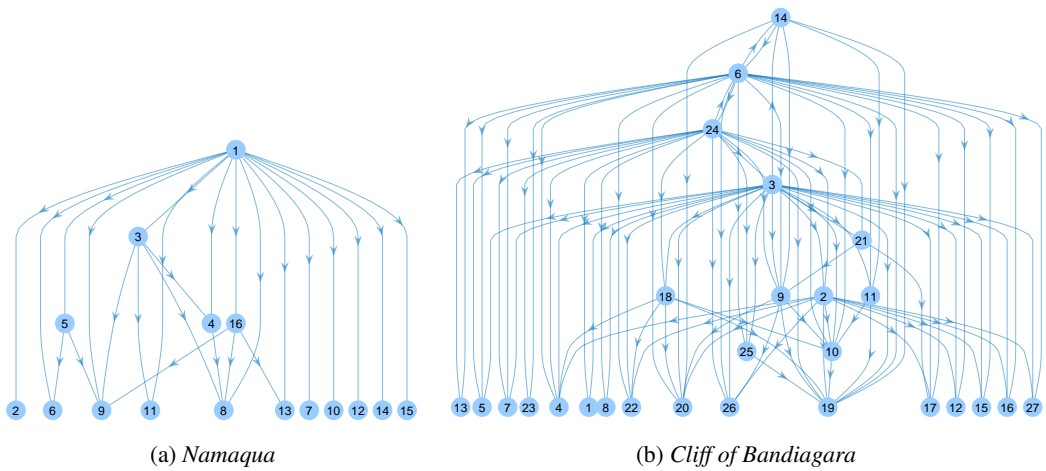

(a) *Namaqua*  (b) *Cliff of Bandiagara*

Figure 22: Examples of food webs as directed graphs.

## A.9 FOOD WEB SPECIES CORRESPONDENCES

Table 2: Species correspondences in *Namaqua* site

| Index | Species Name | Common Name |
|-------|--------------|-------------|
| 1 | Caracal_caracal | Caracal |
| 2 | Proteles_cristata | Aardwolf |
| 3 | Felis_silvestris | Wildcat |
| 4 | Vulpes_chama | Cape fox |
| 5 | Ictonyx_striatus | Striped polecat |
| 6 | Cynictis_penicillata | Yellow mongoose |
| 7 | Herpestes_pulverulentus | Cape gray mongoose |
| 8 | Suricata_suricatta | Meerkat |
| 9 | Sylvicapra_grimmia | Common duiker |
| 10 | Hystrix_africaeaustralis | Cape porcupine |
| 11 | Otocyon_megalotis | Bat eared fox |
| 12 | Lepus_capensis | Cape hare |
| 13 | Lepus_saxatilis | Scrub hare |
| 14 | Genetta_genetta | Common genet |
| 15 | Raphicerus_campestris | Steenbok |
| 16 | Procavia_capensis | Rock hyrax |

Table 3: Species correspondences in *Cliff of Bandiagara* site.

| Index | Species Name | Common Name |
|-------|--------------|-------------|
| 1 | Crocuta_crocuta | Spotted hyena |
| 2 | Panthera_pardus | Leopard |
| 3 | Aonyx_capensis | African clawless otter |
| 4 | Caracal_caracal | Caracal |
| 5 | Leptailurus_serval | Serval |
| 6 | Canis_adustus | Side striped jackal |
| 7 | Mellivora_capensis | Honey badger |
| 8 | Felis_silvestris | Wildcat |
| 9 | Herpestes_ichneumon | Egyptian mongoose |
| 10 | Vulpes_pallida | Pale fox |
| 11 | Ictonyx_libycus | Saharan striped polecat |
| 12 | Mungos_mungo | Banded mongoose |
| 13 | Ictonyx_striatus | Striped polecat |
| 14 | Herpestes_sanguineus | Common slender mongoose |
| 15 | Orycteropus_afer | Aardvark |
| 16 | Hyaena_hyaena | Striped hyena |
| 17 | Civettictis_civetta | African civet |
| 18 | Chlorocebus_sabaeus | Green monkey |
| 19 | Ichneumia_albicauda | White tailed mongoose |
| 20 | Lepus_capensis | Cape hare |
| 21 | Genetta_genetta | Common genet |
| 22 | Lepus_victoriae | African savanna hare |
| 23 | Hippotragus_equinus | Roan antelope |
| 24 | Papio_anubis | Olive baboon |
| 25 | Eudorcas_rufifrons | Red fronted gazelle |
| 26 | Erythrocebus_patas | Common patas monkey |
| 27 | Procavia_capensis | Rock hyrax |

A.10   HYPERPARAMETER ANALYSIS

**Sensitivity Analysis.**   In this section, we conduct sensitivity analyses on the model parameters $(\alpha, \epsilon, \gamma)$, and clarify how they interact.

**The tradeoff parameter $\alpha$.** We set $\alpha = 0.5$ as the default to reflect an equal balance between the zeroth-order (feature-driven) and first-order (structure-driven) terms in Equation equation 1. This choice corresponds to the natural modelling assumption that motif-role similarity and structural similarity contribute comparably to ecological functional equivalence. In Figure 23(a), we found that both the zeroth-order and first-order cost alignment costs are near-constant, and that the first-order cost is larger than the zeroth-order. This justifies $\alpha = 0.5$ as a reasonable trade-off between the respective costs. Moreover, in Figure 23(b), we found that the solution varies smoothly with $\alpha$: perturbing $\alpha$ by $\pm 0.1$ changes the alignment by only $\sim 0.12$ in Frobenius norm. This small and approximately linear response demonstrates that (i) $\alpha = 0.5$ is a robust and interpretable default, and (ii) the method is not sensitive to moderate deviations from this value.

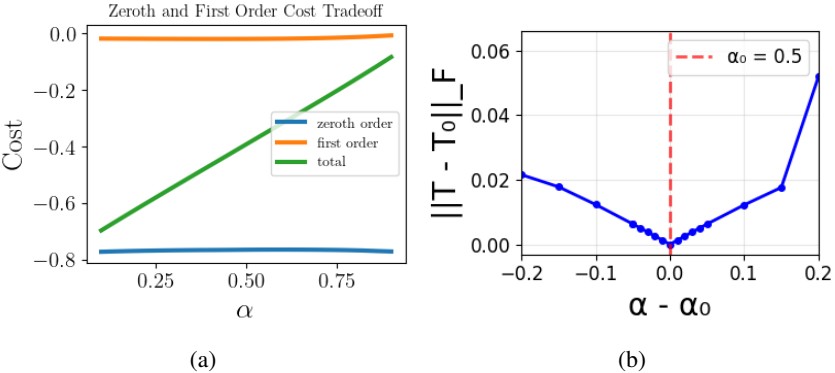

(a)                                              (b)

Figure 23: $\alpha$ Sensitivity

**The self-alignment regularizer $\epsilon$.** We additionally examined how the total transported mass (controlled directly by $\epsilon$) varies with $\epsilon$. For a pair of food webs with sizes $m$ and $n$, respectively, the relationship follows a smooth saturating curve well-approximated by

$$f(\epsilon) = a\big(1 - \exp(-b\epsilon)\big), \tag{6}$$

where $a \approx 0.83$ and $b \approx 4.25(m + n) - 56.5$. Figure 24 shows this approximation on three pairs of real food webs, with a mean $R^2$ value of $0.984$. This indicates that $\epsilon$ influences the degree of self-alignment in a *stable and predictable* manner. This also yields a practical scheme for selecting $\epsilon$ given a desired level of tolerated self-alignment.

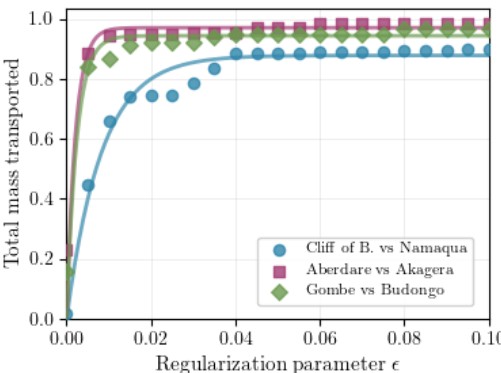

Figure 24: Approximated and empirical total mass transported vs. $\epsilon$ for three pairs of food webs.

**The step size $\gamma$.** In the problem formulation and algorithm, $\gamma$ plays the role of a step-size in the usual sense of a first-order optimization algorithm. We found $\gamma = 0.1$ to be a suitable default value

that balances stability and convergence speed. To test this, we align several pairs of networks, using different values of $\gamma$, and show robustness to the choice of this parameter. In Fig. 25, we show that, when producing an alignment between *Cliff of Bandiagara* and *Namaqua*, our algorithm smoothly minimizes the cost with a $\gamma$-dependent rate of convergence.

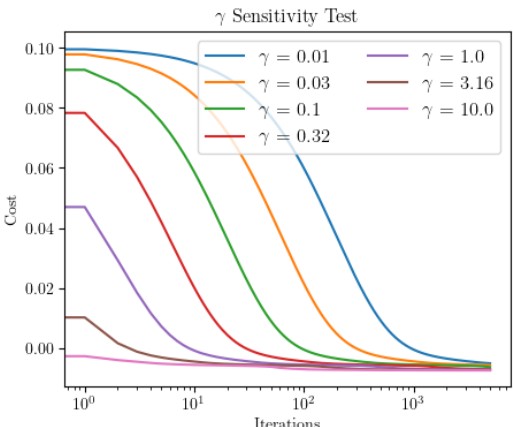

Figure 25: $\gamma$ sensitivity

**Ablation Analysis.**    To further clarify the role of individual terms in the objective, we conducted two ablations:

- **Self-alignment removed** ($\epsilon = 0$)**:** Performance deteriorates sharply. Meta-alignment consistency collapses, and backbones fall below the null model in transitivity, confirming that the self-alignment penalty is essential to avoid trivial low-mass solutions (shown in Figs. 26, 27).

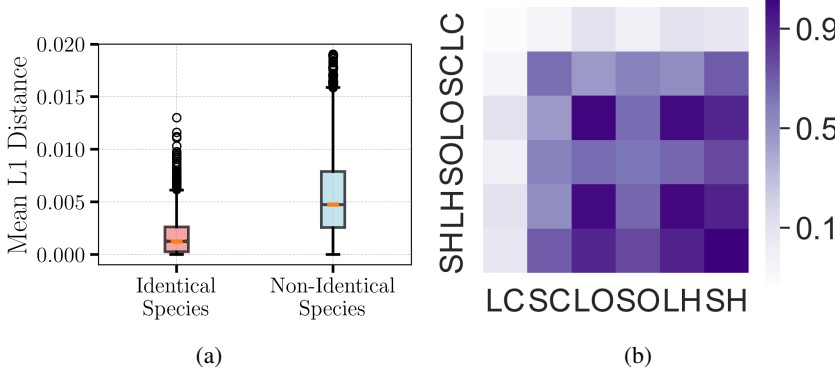

Figure 26: Alignment Discrepancy and Meta-alignment for $\epsilon = 0$

- **First-order term removed** ($\alpha = 0$)**:** Results remain comparable on most tasks, showing that the model is *robust* to the tradeoff parameter. Since $\alpha$ interpolates between two ecologically meaningful similarity sources (motif-based features and structural interactions), $\alpha = 0$ represents a valid special case rather than a failure mode (shown in Figs. 28, 29).

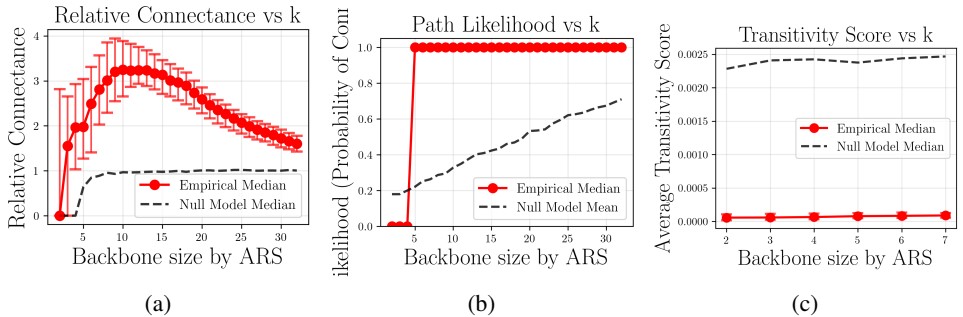

Figure 27: Statistical Analysis of Top-k backbones for $\epsilon = 0$.

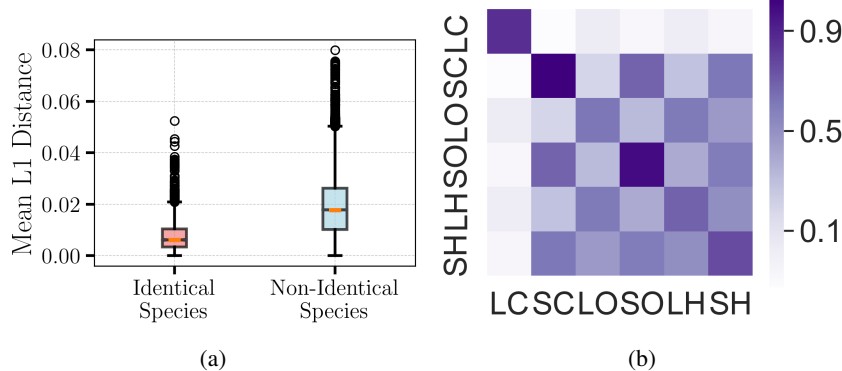

Figure 28: Alignment Discrepancy and Meta-alignment for $\alpha = 0$

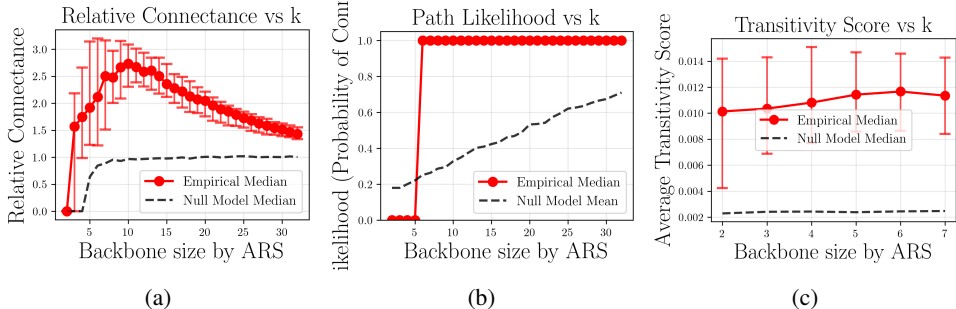

Figure 29: Statistical Analysis of Top-k backbones for $\alpha = 0$

A.11    ALIGNMENTS ON ADDITIONAL REAL AND SYNTHETIC DATA

To demonstrate broader applicability to ecological networks, we also evaluated our method on a subset of food webs from Mora et al. (2018a), for which we obtained directed interaction data. We performed pairwise alignments between selected pairs of food webs, shown in Figure 30. However, the dataset does not include species-level trait profiles or other ecological attributes that would allow for quantitative ecological validation (e.g., comparing aligned species by diet, body size, or functional role). As a result, our evaluation here is restricted to a qualitative inspection of the alignment structure rather than a trait-based assessment.

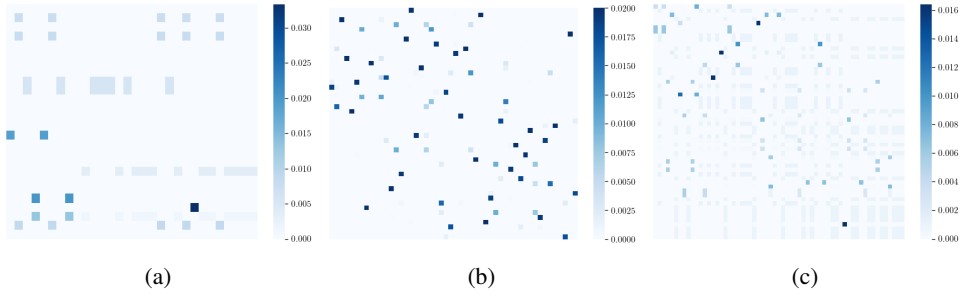

(a)                                    (b)                                    (c)

Figure 30: Examples of pairwise alignments between the dataset used in Mora et al. (2018a). (Left) Coachella (26 species) vs. Connery (30 species); (Middle) Mondego (48 species) vs. Reef (50 species); (Right) Alford (56 species) vs. Beaver (61 species).

To further demonstrate general utility, we additionally aligned synthetic ecological networks generated by the classical Niche model (Williams & Martinez, 2000) and Cascade model (Cohen & Newman, 1985). (Fig. 31, Fig. 32) In both the Niche-model and Cascade-model experiments, we observe that species with similar niche values, which correspond to similar trophic positions in the generative models, tend to align strongly with one another. This is exactly the structural pattern predicted by these models: species with comparable niche parameters have overlapping feeding ranges and play analogous roles in the synthetic food web. The fact that our method reconstructs these relationships from network structure alone, without access to the underlying niche values, demonstrates that the alignment objective captures meaningful functional similarities. This serves as an additional validation step, showing that Wasserstein Motifs recovers ecologically coherent correspondences even in controlled synthetic systems where the ground-truth generative mechanism is known.

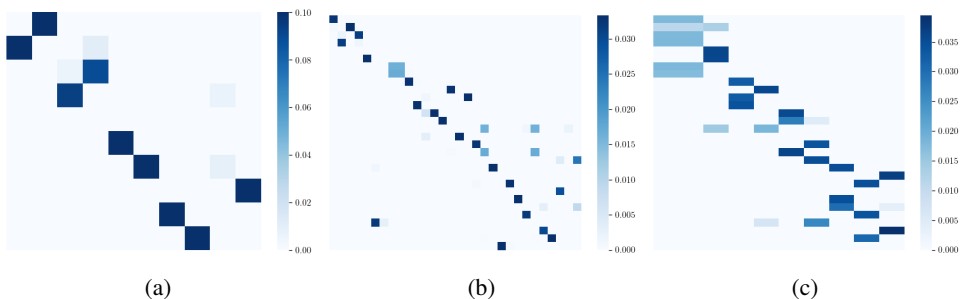

(a)                                    (b)                                    (c)

Figure 31: Examples of pairwise alignments between synthetic food webs generated by the Niche model (Williams & Martinez, 2000). Species are sorted in decreasing niche values, which represent trophic levels. (Left) 10 species, expected connectance 0.4 vs. 10 species, expected connectance 0.3; (Middle) 30 species, expected connectance 0.3 vs. 30 species, expected connectance 0.25; (Right) 30 species, expected connectance 0.3 vs. 10 species, expected connectance 0.4.

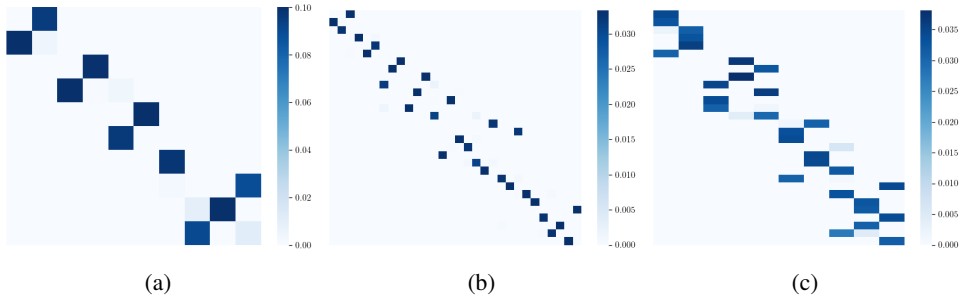

(a)                     (b)                     (c)

Figure 32: Examples of pairwise alignments between synthetic food webs generated by the Cascade model (Cohen & Newman, 1985). Species are sorted in decreasing niche values, which represent trophic levels. (Left) 10 species, expected connectance 0.4 vs. 10 species, expected connectance 0.3; (Middle) 30 species, expected connectance 0.3 vs. 30 species, expected connectance 0.25; (Right) 30 species, expected connectance 0.3 vs. 10 species, expected connectance 0.4.

## B   PROOF OF CLAIMS

### B.1   PROOF OF THEOREM 2.4

We first write the split form of the problem as seen in Equation equation 3:

$$\min_{\pi=w} g(\pi, w) + \mathbb{I}_{\mathcal{C}_1}(\pi) + \mathbb{I}_{\mathcal{C}_2}(w),$$
$$\text{where } \mathcal{C}_1 = \{X \in [0,1]^{m \times n} : X1 \preceq \mu\}, \quad \mathcal{C}_2 = \{X \in [0,1]^{m \times n} : X^\top 1 \preceq \nu\}. \tag{7}$$

Let $h(X) = \sum_{ij} x_{ij} \log x_{ij}$ be the negative entropy function and $D_h$ be its Bregman divergence. Let $\{(\pi^{(k)}, w^{(k)})\}_{k \geq 0}$ be the sequence generated by alternatively solving the subproblems in equation 2. We define the potential function:

$$F_\gamma(\pi, w) \triangleq g(\pi, w) + \mathbb{I}_{\mathcal{C}_1}(\pi) + \mathbb{I}_{\mathcal{C}_2}(w) + \frac{1}{\gamma} D_h(\pi, w).$$

**Lemma B.1.** *The following properties hold:*

1. *$f(\pi, w) = \alpha \langle \pi, A_1(C \odot w)A_2 \rangle$ is bilinear;*

2. *$\mathcal{C}_1$ and $\mathcal{C}_2$ are closed, convex polytopes;*

3. *the potential function $F_\gamma$ is coercive.*

*Proof of Lemma B.1.* The three statements are proved in parallel:

1. This follows immediately due to the linearity of inner products, matrix multiplication, and the Hadamard product.

2. **Closedness:** Both inequality constraints $X1 \preceq \mu$ (resp. $X^\top 1 \preceq \nu$) can be represented as an intersection of closed half-spaces. Since $[0,1]^{m \times n}$ is a product of closed intervals which is also closed, we have that the intersection of closed sets is closed; hence, $\mathcal{C}_1$ and $\mathcal{C}_2$ are closed.
   **Convexity:** Both the unit box $[0,1]^{m \times n}$ and the linear inequalities are convex, and hence their intersection is convex.
   **Polytope:** Both $\mathcal{C}_1$ and $\mathcal{C}_2$ are bounded by the unit box $[0,1]^{m \times n}$. In addition, they are both given by finitely many linear inequalities, so they are polyhedra. Since both are bounded polyhedra, they are polytopes by definition.

3. Since both $\mathcal{C}_1$ and $\mathcal{C}_2$ are bounded, $\mathcal{C}_1 \times \mathcal{C}_2$ is bounded. Hence, whenever $\|(\pi, w)\| \to \infty$, we must have that $(\pi, w) \notin \mathcal{C}_1 \times \mathcal{C}_2$, which implies either $\pi \notin \mathcal{C}_1$ or $w \notin \mathcal{C}_2$. In the former case, we have that $\mathbb{I}_{\mathcal{C}_1}(\pi) \to \infty$, and in the latter case we have that $\mathbb{I}_{\mathcal{C}_2}(w) \to \infty$, both of which implies $F_\gamma(\pi, w) \to \infty$, as desired.

$\square$

*Proof of Theorem 2.4.* Since the accumulative asymmetrical error is bounded and the statements in Lemma B.1 hold, then by Theorem 3.6 in Li et al. (2023), every limit point of the sequence $\{(\pi^{(k)}, w^{(k)})\}_{k \geq 0}$ generated by equation 2 belongs to the fixed point set of the Bregman Alternating Projected Gradient. Hence, the updates equation 2 generate a sequence $\{T^{(k)}\}_{\geq 0}$ that converges to a first-order stationary point of Problem 7, and therefore a first-order stationary point of Equation equation 1. $\square$

### B.2 PROOF OF PROPOSITION 2.2

*Proof of Proposition 2.2.* Let $\mathcal{X}_1 = (X_1, A_1, \nu_{X_1})$ and $\mathcal{X}_2 = (X_2, A_2, \nu_{X_2})$ be measure networks with the uniform discrete measures over their vertex sets. Recall from Equation equation 1 that the feasible set for deterministic alignment is

$$\left\{ T \in \{0,1\}^{m \times n} : T 1_n \preceq \nu_{X_1}, T^\top 1_m \preceq \nu_{X_2} \right\}.$$

Since $\nu_{X_1}$ and $\nu_{X_2}$ each assign one unit of mass to each vertex, the marginal constraints can be further reduced to

$$\sum_{j=1}^{n} T_{ij} \leq 1 \quad \text{and} \quad \sum_{i=1}^{m} T_{ij} \leq 1. \tag{8}$$

Consider the complete bipartite graph $K_{m,n} = (X_1 \cup X_2, E)$ with partition $(X_1, X_2)$. A (not necessarily perfect) matching $\mathcal{M} \subseteq E$ is a set of vertex-disjoint edges.

We now construct an explicit bijection between matchings of $K_{m,n}$ and feasible deterministic alignments.
**Matchings → Deterministic Alignments.** Given a matching $\mathcal{M}$, define $T^{\mathcal{M}} \in \{0,1\}^{m \times n}$ by

$$T_{ij}^{\mathcal{M}} = \begin{cases} 1 & \text{if } (x_i, x_j') \in \mathcal{M}, \\ 0 & \text{otherwise.} \end{cases}$$

Since $\mathcal{M}$ is a matching, no vertex is incident to more than one edge. Hence $T^{\mathcal{M}}$ satisfies equation 8 and is a deterministic alignment.

**Deterministic Alignments → Matchings.** Conversely, let $T$ be a deterministic alignment. Define

$$\mathcal{M}^T = \{(x_i, x_j' \in E : T_{ij} = 1)\}.$$

The marginal constraints in equation 8 imply that each $x_i \in X_1$ and $x_j' \in X_2$ is incident to at most one edge in $\mathcal{M}^T$, making $\mathcal{M}^T$ a valid matching.

**Bijection.** Clearly,

$$\mathcal{M}^{T^{\mathcal{M}}} = \mathcal{M} \quad \text{and} \quad T^{\mathcal{M}^T} = T,$$

which implies that the two maps are inverses of each other. Hence, there is a one-to-one correspondence between matchings of $K_{m,n}$ and deterministic alignments, as desired. □

## C  Bounded accumulative asymmetrical error (AAE): Assumption 2.3

For completeness, we briefly recall the setup we are working on. We work with matrices $T \in \mathbb{R}^{m \times n}$ with strictly positive entries and the entropic generator

$$h(X) = \sum_{i,j} x_{ij} \log x_{ij}, \qquad X = (x_{ij})_{i,j},\ x_{ij} > 0.$$

The associated Bregman divergence is

$$D_h(X \| Y) = \sum_{i,j} \left( x_{ij} \log \frac{x_{ij}}{y_{ij}} - x_{ij} + y_{ij} \right),$$

defined for strictly positive matrices $X, Y$.

We consider the row and column constraint sets

$$C_1(\mu) := \left\{ T \in \mathbb{R}_+^{m \times n} : T 1_n \preceq \mu \right\}, \qquad C_2(\nu) := \left\{ T \in \mathbb{R}_+^{m \times n} : T^\top 1_m \preceq \nu \right\},$$

for fixed vectors $\mu \in \mathbb{R}_+^m$, $\nu \in \mathbb{R}_+^n$, where $1_k$ is the all-ones vector and $\preceq$ denotes component-wise inequality.

We are also going to use the following facts and assumptions:

- $\mu_i > 0$ and $\nu_j > 0$ for all $i, j$, and $\sum_i \mu_i < \infty$, $\sum_j \nu_j < \infty$. We have worked only with positive margins. This is a working assumption that is maintained throughout the paper.

- $C_\varepsilon \in \mathbb{R}^{m \times n}$ and $A_1 \in \mathbb{R}^{m \times m}$, $A_2 \in \mathbb{R}^{n \times n}$ are fixed finite matrices, with $C_\varepsilon := C - \varepsilon\, 1_m 1_n^\top$, for some given $C$ and $\varepsilon \geq 0$.

- The initial iterate $T^{(0)}$ satisfies $T^{(0)} > 0$ and $T^{(0)} \in C_2(\nu)$. This condition holds automatically with our initialization scheme $T^{(0)} = \frac{1}{mn} 1_m 1_n^\top$

We slightly rewrite our proposed algorithm in equation 2 to enhance exposition and clarity of proof. Indexed by $k \geq 1$, equation 2 can be rewriten as:

$$
\begin{aligned}
U_{\mathrm{row}}^{(k)} &:= T^{(k-1)} \odot \exp(-\gamma_k Q^{(k-1)}), \\
\hat{T}^{(k)} &:= P_{C_1(\mu)}\big(U_{\mathrm{row}}^{(k)}\big), \\
U_{\mathrm{col}}^{(k)} &:= \hat{T}^{(k)} \odot \exp(-\gamma_k Q'^{(k)}), \\
T^{(k+1)} &:= P_{C_2(\nu)}\big(U_{\mathrm{col}}^{(k)}\big),
\end{aligned}
\tag{9}
$$

where $P_{C_1(\mu)}$ and $P_{C_2(\nu)}$ denote the KL projections onto $C_1(\mu)$ and $C_2(\nu)$, respectively, and

$$
\begin{aligned}
Q^{(k)} &:= \alpha\, A_1 \big( C_\varepsilon \odot T^{(k-1)} \big) A_2 + \tfrac{1}{2}(1-\alpha)\, C_\varepsilon, \\
Q'^{(k)} &:= \alpha\, C_\varepsilon \odot \big( A_1 \hat{T}^{(k)} A_2 \big) + \tfrac{1}{2}(1-\alpha)\, C_\varepsilon,
\end{aligned}
\tag{10}
$$

for some fixed $\alpha \in [0, 1]$. The step sizes $\gamma_k > 0$ will be specified below.

The multiplicative updates in equation 9 involve exponentials of finite matrices and start from $T^{(0)} > 0$. Thus, they maintain strict positivity at each iteration. Similarly, KL projections onto $C_1(\mu)$ and $C_2(\nu)$ with positive inputs yield strictly positive solutions: the scalar function

$$t \mapsto t \log(t/u) - t + u$$

has derivative $\log(t/u)$, which tends to $-\infty$ as $t \to 0^+$; since the constraints are of the form "$\leq$" with positive right-hand side, the minimizer cannot be at zero. Thus all iterates $T^{(k)}, \hat{T}^{(k)}, U_{\mathrm{row}}^{(k)}, U_{\mathrm{col}}^{(k)}$ have strictly positive entries at any given iteration $k$, yet no uniform lower bound is available.

For the adjacency matrices and cost function, we define the row/column norms:

$$B_1 := \max_{1 \leq i \leq m} \sum_{p=1}^m |(A_1)_{ip}|, \qquad B_2 := \max_{1 \leq j \leq n} \sum_{q=1}^n |(A_2)_{qj}|, \qquad \|C_\varepsilon\|_\infty := \max_{i,j} |(C_\varepsilon)_{ij}|.$$

Finally, we introduce the global bound

$$T_{\max} := \max\{\max_i \mu_i, \max_j \nu_j\} < \infty.$$

**Lemma C.1** (Uniform bound on $Q^{(k)}$ and $Q'^{(k)}$). *Let $Q^{(k)}$ and $Q'^{(k)}$ defined in equation 10, and let*

$$L_Q := \|C_\varepsilon\|_\infty T_{\max}\left(\alpha B_1 B_2 + \frac{1-\alpha}{2}\right).$$

*Thus, for all iterations $k \geq 0$ and all indices $i, j$,*

$$|Q^{(k)}_{ij}| \leq L_Q, \qquad |Q'^{(k)}_{ij}| \leq L_Q.$$

*Proof.* We first bound $Q^{(k)}$. Define

$$M^{(k)} := C_\varepsilon \odot T^{(k-1)}, \qquad M^{(k)}_{pq} = (C_\varepsilon)_{pq} T^{(k-1)}_{pq}.$$

Since $T^{(k-1)} \in C_2(\nu)$, for each column $j$ we have $\sum_i T^{(k-1)}_{ij} \leq \nu_j$, hence $0 < T^{(k-1)}_{ij} \leq \nu_j \leq T_{\max}$. Thus

$$|M^{(k)}_{pq}| \leq \|C_\varepsilon\|_\infty T_{\max} \quad \text{for all } p, q.$$

We then compute

$$(A_1 M^{(k)} A_2)_{ij} = \sum_{p,q} (A_1)_{ip} M^{(k)}_{pq} (A_2)_{qj}.$$

Using the bound on $M^{(k)}$, we obtain

$$\left|(A_1 M^{(k)} A_2)_{ij}\right| \leq \sum_{p,q} |(A_1)_{ip}|\,|M^{(k)}_{pq}|\,|(A_2)_{qj}|$$

$$\leq \|C_\varepsilon\|_\infty T_{\max} \sum_{p,q} |(A_1)_{ip}|\,|(A_2)_{qj}|$$

$$= \|C_\varepsilon\|_\infty T_{\max}\left(\sum_p |(A_1)_{ip}|\right)\left(\sum_q |(A_2)_{qj}|\right)$$

$$\leq \|C_\varepsilon\|_\infty T_{\max} B_1 B_2.$$

The second term in equation 10 satisfies

$$\left|\tfrac{1}{2}(1-\alpha)(C_\varepsilon)_{ij}\right| \leq \tfrac{1}{2}(1-\alpha)\|C_\varepsilon\|_\infty.$$

Therefore,

$$|Q^{(k)}_{ij}| = \left|\alpha(A_1 M^{(k)} A_2)_{ij} + \tfrac{1}{2}(1-\alpha)(C_\varepsilon)_{ij}\right|$$

$$\leq \alpha\|C_\varepsilon\|_\infty T_{\max} B_1 B_2 + \tfrac{1}{2}(1-\alpha)\|C_\varepsilon\|_\infty.$$

By the definition of $L_Q$, we conclude that $|Q^{(k)}_{ij}| \leq L_Q$ for all $i, j, k$.

We now bound $Q'^{(k)}$. Define

$$N^{(k)} := A_1 \hat{T}^{(k)} A_2.$$

Since $\hat{T}^{(k)} \in C_1(\mu)$, for each row $i$ we have $\sum_j \hat{T}^{(k)}_{ij} \leq \mu_i$, hence $0 < \hat{T}^{(k)}_{ij} \leq \mu_i \leq T_{\max}$, and thus

$$|N^{(k)}_{ij}| = \left|\sum_{p,q} (A_1)_{ip} \hat{T}^{(k)}_{pq} (A_2)_{qj}\right|$$

$$\leq \sum_{p,q} |(A_1)_{ip}|\,\hat{T}^{(k)}_{pq}\,|(A_2)_{qj}|$$

$$\leq T_{\max} \sum_{p,q} |(A_1)_{ip}|\,|(A_2)_{qj}|$$

$$\leq T_{\max} B_1 B_2.$$

Hence

$$|(C_\varepsilon \odot N^{(k)})_{ij}| = |(C_\varepsilon)_{ij}|\,|N^{(k)}_{ij}| \leq \|C_\varepsilon\|_\infty T_{\max} B_1 B_2,$$

and therefore

$$|Q'^{(k)}_{ij}| \leq \alpha\|C_\varepsilon\|_\infty T_{\max} B_1 B_2 + \tfrac{1}{2}(1-\alpha)\|C_\varepsilon\|_\infty \leq L_Q.$$

This establishes the claimed bounds for both $Q^{(k)}$ and $Q'^{(k)}$. $\qquad\square$

**Lemma C.2** (Local cubic asymmetry of the scalar entropy). *For scalars $u, v > 0$, define*

$$D(u\|v) := u \log \frac{u}{v} - u + v, \qquad A(u,v) := D(u\|v) - D(v\|u).$$

*Let $\delta_0 = \frac{1}{2}$ and $C_0 = 2$. Then, for any $u, v > 0$ satisfying*

$$|u - v| \leq \delta_0 \min(u, v),$$

*it holds that*

$$|A(u,v)| \leq C_0 \frac{|u-v|^3}{\min(u,v)^2}.$$

*Proof.* We first compute $A(u,v)$ in closed form. From the definition,

$$D(u\|v) = u \log \frac{u}{v} - u + v,$$
$$D(v\|u) = v \log \frac{v}{u} - v + u,$$

so

$$\begin{aligned}
A(u,v) &= D(u\|v) - D(v\|u) \\
&= u \log \frac{u}{v} - u + v - \left( v \log \frac{v}{u} - v + u \right) \\
&= u \log \frac{u}{v} + v \log \frac{u}{v} - 2(u - v) \\
&= (u + v) \log \frac{u}{v} - 2(u - v).
\end{aligned}$$

Without loss of generality, assume $u \geq v$. We may then write $u = rv$ with $r \geq 1$. In that case,

$$u - v = (r - 1)v, \qquad \frac{u}{v} = r,$$

and

$$A(u,v) = v f(r), \qquad f(r) := (r + 1) \log r - 2(r - 1).$$

We now study $f(r)$ near $r = 1$. Direct differentiation yields

$$f(1) = 0, \qquad f'(r) = \log r + \frac{1}{r} - 1, \quad f'(1) = 0,$$

$$f''(r) = \frac{1}{r} - \frac{1}{r^2} = \frac{r - 1}{r^2}, \quad f''(1) = 0,$$

and

$$f^{(3)}(r) = \frac{d}{dr} \left( \frac{r - 1}{r^2} \right) = \frac{2 - r}{r^3}.$$

The condition $|u - v| \leq \delta_0 \min(u, v)$ with $\delta_0 = 1/2$ and $u \geq v$ becomes

$$|u - v| = (r - 1)v \leq \frac{1}{2}v \quad \Rightarrow \quad |r - 1| \leq \frac{1}{2},$$

that is, $r \in [1/2, 3/2]$. For such $r$,

$$|f^{(3)}(r)| = \frac{|2 - r|}{r^3} \leq \frac{2 - 1/2}{(1/2)^3} = \frac{3/2}{1/8} = 12.$$

Let $M_3 := 12$.

By Taylor's theorem with Lagrange remainder around $r = 1$, there exists a point $\xi_r$ between 1 and $r$ such that

$$f(r) = f(1) + f'(1)(r - 1) + \frac{1}{2} f''(1)(r - 1)^2 + \frac{1}{6} f^{(3)}(\xi_r)(r - 1)^3.$$

Since $f(1) = f'(1) = f''(1) = 0$, this simplifies to

$$f(r) = \frac{1}{6} f^{(3)}(\xi_r)(r - 1)^3.$$

Therefore, for all $r$ with $|r - 1| \leq 1/2$,

$$|f(r)| \leq \frac{M_3}{6}|r - 1|^3 = 2|r - 1|^3.$$

Recalling that $r = u/v$ and $\min(u, v) = v$ in the case $u \geq v$, we note that

$$|r - 1| = \left|\frac{u}{v} - 1\right| = \frac{|u - v|}{v} = \frac{|u - v|}{\min(u, v)}.$$

It follows that

$$\begin{aligned}|A(u, v)| &= v|f(r)| \\ &\leq 2v|r - 1|^3 \\ &= 2v\left(\frac{|u - v|}{v}\right)^3 \\ &= 2\frac{|u - v|^3}{v^2} \\ &= 2\frac{|u - v|^3}{\min(u, v)^2}.\end{aligned}$$

This proves the desired inequality when $u \geq v$.

If $u < v$, then the condition $|u - v| \leq \delta_0 \min(u, v)$ is symmetric in $(u, v)$, and $A(u, v) = -A(v, u)$. Hence, the same bound holds in this case as well. The lemma is therefore proved with $\delta_0 = \frac{1}{2}$ and $C_0 = 2$. □

*Remark* C.3. For matrices $X = (x_{ij})$ and $Y = (y_{ij})$ with strictly positive entries, if

$$|x_{ij} - y_{ij}| \leq \delta_0 \min(x_{ij}, y_{ij}) \quad \forall i, j,$$

then, with $D$ and $A$ as in Lemma C.2,

$$D_h(X\|Y) - D_h(Y\|X) = \sum_{i,j} A(x_{ij}, y_{ij}),$$

and Lemma C.2 gives

$$\left|D_h(X\|Y) - D_h(Y\|X)\right| \leq C_0 \sum_{i,j} \frac{|x_{ij} - y_{ij}|^3}{\min(x_{ij}, y_{ij})^2}.$$

**Lemma C.4** (Marginal stability of the $C_2(\nu)$ block)**.** *Let $L_Q$ be as in Lemma C.1, and suppose the iterates $(T^{(k)}, \hat{T}^{(k)})$ are generated by the scheme equation 9–equation 10 with strictly positive initial point $T^{(0)} \in C_2(\nu)$. Assume the step sizes satisfy*

$$0 < \gamma_k \leq \frac{1}{4L_Q} \quad \text{for all } k.$$

*Then there exists a constant*

$$C_g := 40L_Q$$

*such that, for all $k \geq 0$ and all indices $i, j$,*

$$\left|T_{ij}^{(k+1)} - \hat{T}_{ij}^{(k)}\right| \leq C_g\,\gamma_k\,\min\left(T_{ij}^{(k+1)}, \hat{T}_{ij}^{(k)}\right).$$

*Proof.* Fix an iteration index $k \geq 1$ and, to simplify notation, denote

$$T^- := T^{(k-1)}, \quad U_{\text{row}} := U_{\text{row}}^{(k)}, \quad \hat{T} := \hat{T}^{(k)}, \quad U_{\text{col}} := U_{\text{col}}^{(k)}, \quad T^+ := T^{(k+1)},$$

and

$$Q := Q^{(k-1)}, \quad Q' := Q'^{(k)}, \quad \gamma := \gamma_k.$$

By construction, $T^-, T^+ \in C_2(\nu)$ and all matrices have strictly positive entries. Lemma C.1 gives $|Q_{ij}| \leq L_Q$ and $|Q'_{ij}| \leq L_Q$.

We first control the column sums after the row update. The multiplicative step is given by

$$U_{\text{row},ij} = T_{ij}^- e^{-\gamma Q_{ij}}.$$

For each column $j$,

$$S_j(U_{\text{row}}) := \sum_i U_{\text{row},ij} = \sum_i T_{ij}^- e^{-\gamma Q_{ij}}$$

$$\leq e^{\gamma L_Q} \sum_i T_{ij}^- \leq e^{\gamma L_Q} \nu_j,$$

where we used $|Q_{ij}| \leq L_Q$ and $T^- \in C_2(\nu)$.

Next, $\hat{T}$ is the KL projection of $U_{\text{row}}$ onto $C_1(\mu)$, which is row-wise. For each row $i$, the projection problem is

$$\min_{t_{i\cdot} \geq 0} \sum_j \left( t_{ij} \log \frac{t_{ij}}{u_{ij}} - t_{ij} + u_{ij} \right) \quad \text{s.t.} \quad \sum_j t_{ij} \leq \mu_i,$$

where $u_{ij} := U_{\text{row},ij}$. The KKT conditions show that

$$\hat{T}_{ij} = \alpha_i U_{\text{row},ij}, \qquad \alpha_i := \min\left(1, \frac{\mu_i}{\sum_\ell U_{\text{row},i\ell}}\right) \in (0,1].$$

In particular, $\hat{T}_{ij} \leq U_{\text{row},ij}$ for all $i,j$. Consequently, for each column $j$,

$$S_j(\hat{T}) := \sum_i \hat{T}_{ij} \leq \sum_i U_{\text{row},ij} = S_j(U_{\text{row}}) \leq e^{\gamma L_Q} \nu_j.$$

We now consider the second multiplicative step

$$U_{\text{col},ij} = \hat{T}_{ij} e^{-\gamma Q'_{ij}}.$$

For each column $j$,

$$S_j(U_{\text{col}}) := \sum_i U_{\text{col},ij} = \sum_i \hat{T}_{ij} e^{-\gamma Q'_{ij}}$$

$$\leq e^{\gamma L_Q} \sum_i \hat{T}_{ij}$$

$$\leq e^{2\gamma L_Q} \nu_j,$$

using $|Q'_{ij}| \leq L_Q$ and the bound on $S_j(\hat{T})$.

For $x \in [0,1]$, the function $g(x) := 1 + 2x - e^x$ satisfies $g(0) = 0$ and $g'(x) = 2 - e^x$, hence $g(x) \geq 0$ on $[0,1]$. Thus $e^x \leq 1 + 2x$ whenever $x \in [0,1]$. Since $\gamma L_Q \leq 1/(4L_Q) \cdot L_Q = 1/4$, we have $2\gamma L_Q \leq 1/2 < 1$ and therefore

$$e^{2\gamma L_Q} \leq 1 + 4\gamma L_Q.$$

It follows that

$$S_j(U_{\text{col}}) \leq (1 + 4\gamma L_Q)\nu_j \quad \text{for all } j.$$

We now examine the KL projection of $U_{\text{col}}$ onto $C_2(\nu)$. For a fixed column $j$, we solve

$$\min_{t_{\cdot j} \geq 0} \sum_i \left( t_{ij} \log \frac{t_{ij}}{u_{ij}} - t_{ij} + u_{ij} \right) \quad \text{s.t.} \quad \sum_i t_{ij} \leq \nu_j,$$

where $u_{ij} := U_{\text{col},ij}$. The Lagrangian is

$$\mathcal{L}(t,\lambda) = \sum_i \left( t_{ij} \log \frac{t_{ij}}{u_{ij}} - t_{ij} + u_{ij} \right) + \lambda \left( \sum_i t_{ij} - \nu_j \right), \quad \lambda \geq 0.$$

First-order optimality for $t_{ij} > 0$ yields

$$\frac{\partial \mathcal{L}}{\partial t_{ij}} = \log \frac{t_{ij}}{u_{ij}} + \lambda = 0 \quad \Rightarrow \quad t_{ij} = u_{ij} e^{-\lambda}.$$

Thus the minimizer has the form

$$T_{ij}^+ = \beta_j U_{\text{col},ij}, \quad \text{with} \quad \beta_j := e^{-\lambda} > 0.$$

The complementary slackness and feasibility conditions imply

$$\sum_i T_{ij}^+ \leq \nu_j, \quad \lambda \geq 0, \quad \lambda\Big(\sum_i T_{ij}^+ - \nu_j\Big) = 0.$$

Hence either the constraint is inactive and $\beta_j = 1$, or the constraint is active and

$$\sum_i T_{ij}^+ = \beta_j \sum_i U_{\text{col},ij} = \nu_j \quad \Rightarrow \quad \beta_j = \frac{\nu_j}{S_j(U_{\text{col}})}.$$

In all cases,

$$T_{ij}^+ = \beta_j U_{\text{col},ij}, \quad \beta_j = \min\left(1, \frac{\nu_j}{S_j(U_{\text{col}})}\right).$$

If $S_j(U_{\text{col}}) \leq \nu_j$, then $\beta_j = 1$ and $T_{ij}^+ = U_{\text{col},ij}$. If $S_j(U_{\text{col}}) > \nu_j$, then the bound $S_j(U_{\text{col}}) \leq (1 + 4\gamma L_Q)\nu_j$ implies

$$\beta_j = \frac{\nu_j}{S_j(U_{\text{col}})} \geq \frac{\nu_j}{(1 + 4\gamma L_Q)\nu_j} = \frac{1}{1 + 4\gamma L_Q}.$$

In that case,

$$1 - \beta_j = 1 - \frac{\nu_j}{S_j(U_{\text{col}})} = \frac{S_j(U_{\text{col}}) - \nu_j}{S_j(U_{\text{col}})} \leq \frac{(1 + 4\gamma L_Q)\nu_j - \nu_j}{\nu_j} = 4\gamma L_Q.$$

Consequently, in both cases, the bound

$$|1 - \beta_j| \leq 4\gamma L_Q$$

holds. Moreover, if $\gamma \leq 1/(4L_Q)$, then $4\gamma L_Q \leq 1$ and

$$\beta_j \geq \frac{1}{1 + 4\gamma L_Q} \geq \frac{1}{2}.$$

Using $T_{ij}^+ = \beta_j U_{\text{col},ij}$, we deduce

$$|T_{ij}^+ - U_{\text{col},ij}| = U_{\text{col},ij}|1 - \beta_j| \leq 4L_Q\gamma\, U_{\text{col},ij}.$$

On the other hand, $\beta_j \in [1/2, 1]$ implies $T_{ij}^+ \leq U_{\text{col},ij}$ and $T_{ij}^+ \geq \frac{1}{2}U_{\text{col},ij}$, whence

$$\min(T_{ij}^+, U_{\text{col},ij}) = T_{ij}^+ \geq \frac{1}{2}U_{\text{col},ij},$$

and the previous inequality may also be written as

$$|T_{ij}^+ - U_{\text{col},ij}| \leq 8L_Q\gamma\, \min(T_{ij}^+, U_{\text{col},ij}).$$

We now relate $U_{\text{col}}$ and $\hat{T}$. From the definition,

$$U_{\text{col},ij} = \hat{T}_{ij}\exp(-\gamma Q'_{ij}),$$

and by Lemma C.1, $|Q'_{ij}| \leq L_Q$. Thus $|\gamma Q'_{ij}| \leq \gamma L_Q \leq 1/4$. For any $x$ with $|x| \leq 1/2$, the function $e^{-x}$ satisfies $|e^{-x} - 1| \leq e^{|x|}|x| \leq e^{1/2}|x| < 2|x|$, hence

$$|e^{-\gamma Q'_{ij}} - 1| \leq 2|\gamma Q'_{ij}| \leq 2\gamma L_Q.$$

This yields

$$|U_{\text{col},ij} - \hat{T}_{ij}| = \hat{T}_{ij}|e^{-\gamma Q'_{ij}} - 1| \leq 2L_Q\gamma\, \hat{T}_{ij}.$$

Furthermore, since $|\gamma Q'_{ij}| \leq 1/4$, we have $e^{-1/4} \leq e^{-\gamma Q'_{ij}} \leq e^{1/4}$, and the numerical estimates $e^{1/4} < 2$ and $e^{-1/4} > 1/2$ imply

$$\frac{1}{2}\hat{T}_{ij} \leq U_{\text{col},ij} \leq 2\hat{T}_{ij}.$$

Introduce the quantity
$$m_{ij} := \min(T_{ij}^+, \hat{T}_{ij}).$$

From the inequality $U_{\text{col},ij} \geq \frac{1}{2}\hat{T}_{ij}$ and the bound $T_{ij}^+ \geq \frac{1}{2}U_{\text{col},ij}$, we obtain

$$T_{ij}^+ \geq \frac{1}{2}U_{\text{col},ij} \geq \frac{1}{2} \cdot \frac{1}{2}\hat{T}_{ij} = \frac{1}{4}\hat{T}_{ij}.$$

Thus $\hat{T}_{ij} \leq 4T_{ij}^+$. Combining this with $U_{\text{col},ij} \leq 2\hat{T}_{ij}$ yields $U_{\text{col},ij} \leq 8T_{ij}^+$.

If $m_{ij} = \hat{T}_{ij} \leq T_{ij}^+$, then $\hat{T}_{ij} = m_{ij}$ and $U_{\text{col},ij} \leq 2\hat{T}_{ij} = 2m_{ij}$. If $m_{ij} = T_{ij}^+ < \hat{T}_{ij}$, then $\hat{T}_{ij} \leq 4T_{ij}^+ = 4m_{ij}$ and $U_{\text{col},ij} \leq 8T_{ij}^+ = 8m_{ij}$. In all cases,

$$\hat{T}_{ij} \leq 4m_{ij}, \qquad U_{\text{col},ij} \leq 8m_{ij}.$$

We now combine the previous inequalities:

$$\begin{aligned}
|T_{ij}^+ - \hat{T}_{ij}| &\leq |T_{ij}^+ - U_{\text{col},ij}| + |U_{\text{col},ij} - \hat{T}_{ij}| \\
&\leq 4L_Q\gamma\, U_{\text{col},ij} + 2L_Q\gamma\, \hat{T}_{ij} \\
&\leq 4L_Q\gamma \cdot 8m_{ij} + 2L_Q\gamma \cdot 4m_{ij} \\
&= (32 + 8)L_Q\gamma\, m_{ij} \\
&= 40L_Q\gamma\, \min(T_{ij}^+, \hat{T}_{ij}).
\end{aligned}$$

Defining $C_g := 40L_Q$ gives the desired inequality

$$|T_{ij}^+ - \hat{T}_{ij}| \leq C_g\gamma\, \min(T_{ij}^+, \hat{T}_{ij}).$$

Since the argument holds for every $k \geq 1$ and all $i, j$, the lemma is proved. $\qquad\square$

**Theorem C.5** (Summability of the asymmetric error for entropic $h$). *Let $h(X) = \sum_{i,j} x_{ij} \log x_{ij}$ and $D_h$ be its associated Bregman divergence. Let $(T^{(k)}, \hat{T}^{(k)})$ be the sequence generated by the scheme equation 9–equation 10 under the setup and assumptions stated above. Define, for each $k \geq 0$,*
$$\Delta_k := D_h\big(\hat{T}^{(k)}, T^{(k+1)}\big) - D_h\big(T^{(k+1)}, \hat{T}^{(k)}\big).$$
*Assume the step sizes satisfy*

$$0 < \gamma_k \leq \bar{\gamma} := \frac{1}{80L_Q} \quad \text{for all } k, \qquad \sum_{k=0}^{\infty} \gamma_k^3 < \infty.$$

*Then*

$$\sum_{k=0}^{\infty} |\Delta_k| < \infty.$$

*In particular, the accumulative asymmetric error*

$$\sum_{k=0}^{\infty} \big(D_h(\hat{T}^{(k)}, T^{(k+1)}) - D_h(T^{(k+1)}, \hat{T}^{(k)})\big)$$

*is absolutely convergent, i.e., Assumption 1 holds for the entropic generator $h$ under these conditions.*

*Proof.* Since $\hat{T}^{(k)} \in C_1(\mu)$ for all $k$, we have

$$\sum_{i,j} \hat{T}_{ij}^{(k)} = \sum_i (\hat{T}^{(k)} 1_n)_i \leq \sum_i \mu_i =: M < \infty.$$

By Lemma C.4 with $C_g = 40L_Q$, for each $k, i, j$,

$$|T_{ij}^{(k+1)} - \hat{T}_{ij}^{(k)}| \leq C_g\gamma_k\, \min\big(T_{ij}^{(k+1)}, \hat{T}_{ij}^{(k)}\big).$$

Since $\gamma_k \leq \bar{\gamma} = 1/(80L_Q)$, we obtain

$$C_g \gamma_k = 40L_Q \gamma_k \leq 40L_Q \cdot \frac{1}{80L_Q} = \frac{1}{2}.$$

Let $\delta_0 = \frac{1}{2}$ be the constant from Lemma C.2. Then

$$\left| T_{ij}^{(k+1)} - \hat{T}_{ij}^{(k)} \right| \leq \delta_0 \min\left( T_{ij}^{(k+1)}, \hat{T}_{ij}^{(k)} \right) \quad \text{for all } i, j, k.$$

Define

$$u_{ij}^{(k)} := \hat{T}_{ij}^{(k)}, \qquad v_{ij}^{(k)} := T_{ij}^{(k+1)}, \qquad m_{ij}^{(k)} := \min(u_{ij}^{(k)}, v_{ij}^{(k)}).$$

The inequality above shows that the pair $(u_{ij}^{(k)}, v_{ij}^{(k)})$ satisfies the locality condition of Lemma C.2 for all $i, j, k$. Hence Lemma C.2 gives

$$\left| D(u_{ij}^{(k)} \| v_{ij}^{(k)}) - D(v_{ij}^{(k)} \| u_{ij}^{(k)}) \right| \leq C_0 \frac{|u_{ij}^{(k)} - v_{ij}^{(k)}|^3}{(m_{ij}^{(k)})^2},$$

where $C_0 = 2$. Using Lemma C.4 again,

$$|u_{ij}^{(k)} - v_{ij}^{(k)}| = |\hat{T}_{ij}^{(k)} - T_{ij}^{(k+1)}| \leq C_g \gamma_k \, m_{ij}^{(k)}.$$

It follows that

$$\frac{|u_{ij}^{(k)} - v_{ij}^{(k)}|^3}{(m_{ij}^{(k)})^2} \leq C_g^3 \gamma_k^3 m_{ij}^{(k)}.$$

Therefore,

$$\left| D(u_{ij}^{(k)} \| v_{ij}^{(k)}) - D(v_{ij}^{(k)} \| u_{ij}^{(k)}) \right| \leq C_0 C_g^3 \gamma_k^3 m_{ij}^{(k)}.$$

The matrix-level asymmetric error can be written as

$$\Delta_k = D_h(\hat{T}^{(k)}, T^{(k+1)}) - D_h(T^{(k+1)}, \hat{T}^{(k)}) = \sum_{i,j} \left( D(u_{ij}^{(k)} \| v_{ij}^{(k)}) - D(v_{ij}^{(k)} \| u_{ij}^{(k)}) \right).$$

Taking absolute values and applying the previous bound yields

$$|\Delta_k| \leq \sum_{i,j} \left| D(u_{ij}^{(k)} \| v_{ij}^{(k)}) - D(v_{ij}^{(k)} \| u_{ij}^{(k)}) \right| \leq C_0 C_g^3 \gamma_k^3 \sum_{i,j} m_{ij}^{(k)}.$$

Since $m_{ij}^{(k)} \leq u_{ij}^{(k)} = \hat{T}_{ij}^{(k)}$, we have

$$\sum_{i,j} m_{ij}^{(k)} \leq \sum_{i,j} \hat{T}_{ij}^{(k)} \leq M,$$

where $M := \sum_i \mu_i < \infty$. Thus

$$|\Delta_k| \leq C_* \gamma_k^3, \qquad C_* := C_0 C_g^3 M.$$

Finally, by the assumption $\sum_{k=0}^\infty \gamma_k^3 < \infty$, we obtain

$$\sum_{k=0}^\infty |\Delta_k| \leq C_* \sum_{k=0}^\infty \gamma_k^3 < \infty.$$

This establishes the claimed absolute convergence of the series $\sum_k |\Delta_k|$, and hence Assumption 1 holds for the entropic generator $h$ under the stated conditions. $\qquad\square$

