# OpenReview forum: "Wasserstein Motifs: Non-deterministic Alignment of Ecological Networks"
_ICLR.cc/2026/Workshop/LMRL — ICLR 2026 Workshop LMRL Poster_

### Official Review · Reviewer_gmQf · 2026-02-20
**Review of 'Wasserstein Motifs'**

**Rating:** 5
**Confidence:** 2

**Review:**

This paper suggests a way to organise ecological food webs by combining features of nodes with a Fused Gromov-Wasserstein-style objective, solved via a KL-Bregman alternating projected gradient scheme. The main idea is well-explained and the work is generally very careful.

The theoretical contributions are solid: the equivalence between Mora et al.'s cost and a special case of the proposed objective is a non-trivial result, and the convergence analysis is thorough and technically credible.

My main concern is validation. The absence of species-level functional ground truth is acknowledged, but the proxy metrics used (alignment discrepancy for identical vs. non-identical species, backbone connectivity/transitivity) largely test structural self-consistency rather than ecological validity. The single ecologically interpretable case study, the mongoose/steenbok alignment, is compelling but anecdotal.
The baseline comparison is also somewhat stacked: FGW and Partial-FGW require trait features that are not always available, placing them at a disadvantage in the chosen experimental setting. A controlled comparison where traits are available for all methods would strengthen the claims about the objective design being the driver of improvement

A marginal but noteworthy issue concerns the manuscript's layout. Several figures appear displaced relative to their in-text references, and the overall figure placement, particularly in the appendix, suggests the template may have been modified to accommodate space constraints. While this does not affect the scientific content, it hinders readability and raises questions about the care taken in the final preparation of the submission.

---

### Official Review · Reviewer_6dvx · 2026-02-20

**Rating:** 8
**Confidence:** 4

**Review:**

This paper presents "Wasserstein Motifs," an innovative mathematical framework designed for the non-deterministic alignment of ecological networks, like food webs. By creating a Fused Gromov-Wasserstein-like objective centered around network motifs, the authors tackle the inflexibility and scalability issues that come with traditional deterministic alignment methods. They introduce a streamlined KL Bregman Alternating Projected Gradient (KL-BAPG) algorithm that comes with convergence guarantees. To demonstrate the effectiveness of their approach, they apply it to a large-scale dataset of mammal food webs from Sub-Saharan Africa, successfully extracting robust non-deterministic "backbones of interactions."

Quality:

The technical execution of this paper is exceptional.

1. The authors take a deep dive into the ecological network alignment problem by applying optimal transport methods. They ease the strict bijection requirement found in the Quadratic Assignment Problem (QAP), allowing for a more flexible many-to-many non-deterministic alignment.

2. Their proposed objective function strikes a thoughtful balance between zeroth-order feature similarity, using motif role profiles, and first-order structural similarity.

3. To tackle this non-convex challenge, the authors implement a KL-BAPG scheme with Dykstra projections and back it up with a robust theoretical proof demonstrating its convergence to first-order stationary points, as long as the cumulative asymmetrical error remains within bounds.

4. The empirical validation is quite thorough, working with a huge dataset that includes 129 food webs from Sub-Saharan African mammals, featuring 216 unique species. This method boasts an impressive 40 times faster performance compared to the simulated annealing baseline set by Mora et al. (2018a), reducing the alignment time to about $0.20 \pm 0.08$ seconds.

Clarity:

The paper is exceptionally clear and structured logically.

1. The authors do a fantastic job of guiding readers through the complex world of feature measure networks. They make sure that moving from discrete ecological motifs to continuous optimization landscapes feels smooth and straightforward.

2. The visual diagrams included in the paper, especially the illustrations of bipartite network alignment, really highlight the benefits of non-deterministic alignment compared to older methods like vanilla Gromov-Wasserstein or rigid deterministic matching. Plus, the formal definitions for ecological tasks, like what "top-k backbones" and the "non-deterministic transitivity score" (Def. 4.3) mean, are laid out with great precision.

Originality:

The work showcases a high level of originality. While network motifs and Gromov-Wasserstein distances are commonly used in representation learning, the way they are combined to model partial and non-deterministic alignments in ecology is quite innovative. Previous motif-based approaches imposed rigid one-to-one species mappings, completely overlooking the biological reality of functional redundancy in ecosystems. Additionally, traditional optimal transport methods strictly adhere to mass conservation, which can lead to misleading alignments. The introduction of the shifted cost matrix, along with a self-alignment penalty $\epsilon$ to handle unaligned mass, represents a clever and original contribution to the algorithmic landscape.

Significance:

The importance of the LMRL workshop for the wider computational ecology community is huge. One of the major challenges in biodiversity research is mapping the functional roles of species across vastly different geographic areas without using direct node labels. By offering a tool that is scalable, mathematically sound, and ecologically meaningful, the authors are paving the way for large-scale ecosystem analysis that was once too complex to tackle.

Pros:

1. Scalability: The KL-BAPG algorithm achieves an impressive 40x reduction in runtime when compared to current ecological network alignment methods.

2. Ecological Realism: This approach supports non-deterministic (many-to-many) correspondences and self-alignment, which allows it to naturally represent functional redundancy in food webs.

3. Strong Theoretical Grounding: The convergence of the optimization scheme is thoroughly proven.

4. Thorough Validation: It has been tested on a substantial and highly relevant dataset of 129 African mammal food webs, demonstrating excellent coherence and backbone transitivity.

Cons:

1. Undirected Adjacency Reliance: Although this method employs motifs that capture the direction of edges, the first-order structural term actually depends on a symmetrized (undirected) adjacency matrix. The authors point out in Appendix A.6 that using strictly directed adjacencies can hurt meta-alignment performance and backbone transitivity, which poses a slight limitation for tracking purely directed interactions.

2. Inability to natively fuse external traits: This method focuses on structural motif profiles as the main feature descriptor. While it works well, it doesn’t have a clear way to combine node features (like continuous body mass) within the optimal transport cost without reverting to the much slower Fused Gromov-Wasserstein baselines discussed in the appendix.

---

### Official Review · Reviewer_xYg4 · 2026-02-24
**Optimal Transport Framework for Ecological Network Alignment**

**Rating:** 7
**Confidence:** 2

**Review:**

This paper introduces Wasserstein Motifs, a motif-based, optimal-transport-inspired framework for non-deterministic (many-to-many) alignment of ecological networks (food webs), with an associated method for extracting non-deterministic “backbones” of conserved interactions across ecosystems. It formalizes motif-based ecological alignment and shows connections between prior motif-alignment approaches and a Fused Gromov–Wasserstein-like objective, then proposes an efficient KL-Bregman Alternating Projected Gradient (KL-BAPG) solver with convergence-to-stationary-point guarantees under standard assumptions. Experiments on 129 Sub-Saharan African mammal food webs show improvements in alignment coherence, runtime, and backbone properties compared to ecological baselines and OT baselines.
The paper is generally clear and well structured: motivation → formalization → algorithm → experiments → backbone construction and the figures are helpful.

The authors provide a solid mathematical foundation for their method, formulating the alignment problem as a minimization of a Fused Gromov-Wasserstein-like objective. The derivation of the KL-BAPG algorithm is non-trivial and supported by convergence guarantees (Theorem 2.3), which is a strong theoretical contribution. The main limitation to significance is external validity: results are compelling on the given mammal food-web dataset, but it’s not yet fully demonstrated how broadly the approach transfers to other taxa, interaction types, or noisier/partially observed webs.

Where clarity could improve: the cost functional and “why this is the right motif-structural coupling” takes effort to parse on first read, and the relationship to FGW/GW is subtle. The appendix helps, but the main text might benefit from a short “intuition box” explaining what information the first-order term captures that GW/FGW misses.
While OT for graph/network alignment is not new, the specific objective structure, ecological motif integration, and backbone pipeline feel meaningfully new and well motivated.

**Pros:**
-  Strong problem framing and ecological motivation for non-deterministic, potentially partial alignments (functional redundancy, unique roles).
- Technically grounded formulation: explicit objective, constraints, and role of the self-alignment penalty.
- Scalable algorithmic contribution with a convergence-to-stationary-point result under reasonable conditions.
- Interpretability: motif role profiles connect naturally to ecological “roles”; alignment visualizations grouped by diet/biomass are convincing.
- Backbone methodology is a real add-on (not just alignment), with thoughtful metrics including a fractional transitivity definition and null-model comparisons.
- Large-scale empirical study: thousands of pairwise alignments across 129 sites; runtime discussion identifies realistic bottlenecks (motif enumeration/cost construction.

**Cons**

- No ground-truth alignment labels, so “accuracy” is supported indirectly; conclusions depend on proxies (identical species consistency, dietary/biomass coherence, backbone metrics).
- Motif computation bottleneck: alignment solver is fast, but motif enumeration dominates; scalability to larger webs or higher-order motifs may be constrained.
- Directedness handling is somewhat unsatisfying: direction is moved into motif profiles while adjacency is symmetrized; they note directed adjacencies performed worse (Appendix reference), but this is an important modeling compromise for food webs.
- Hyperparameter selection: α fixed to 0.5 by default and ϵ chosen via an approximation function; while sensitivity is in the appendix, the main text could give clearer practical guidance.
- Stationary-point guarantee (not global optimum): fine for nonconvex OT-like objectives, but results could be initialization/step-size sensitive; more discussion of stability across runs would help.
- Baseline comparability: they argue Partial-FGW is infeasible at scale and discuss variants; still, readers may want clearer apples-to-apples settings and a concise summary table of what inputs each method uses (motifs vs traits vs topology).

---

### Meta-Review · Area_Chair_CcqS · 2026-02-28

**Recommendation:** Accept (Poster)
**Confidence:** 4

**Metareview:**

Accept

---

### Decision · Program_Chairs · 2026-03-02

**Decision:**

Accept (Oral)

**Comment:**

Please see the meta-review.